# Transient non-soluble noble metal transport in hydrothermal ore systems

Néstor Cano [1,2] ✉, José M. González-Jiménez [3], Antoni Camprubí [1,2], Eric Morales-Casique [1] & Eduardo González-Partida [4]

The transport of noble metals (Au, Ag) by metal-rich melts in hydrothermal ore systems is now acknowledged as a complementary mechanism to complexing ligands in solution. However, it is unclear where/when both mechanisms coexist and whether metal-rich melts can be physically transported by hydrothermal fluids. Here we show evidence for a suspension-like transport of nano-to-micron-sized metal-rich sulfide-sulfosalt melts within epithermal fluids at <400 °C, forming irregular and bleb-like polymineral inclusions of Ag-Au-Cu-Pb-(-Fe-Zn)-As-Sb-S-Se upon cooling. These polymineral inclusions, 5 nm to 40 μm in size, are cogenetic with fluid inclusions in quartz. Numerical modeling based on particle fluidization and settling theory shows hydrothermal fluids can mechanically transport metal-rich sulfide-sulfosalt nano-micromelts at fluid flow rates $<10^{-1}$ m/s. The chemical similarity between nano- and micron-scale polymineral inclusions suggests the coalescence of nanomelt precursors during transient transport from their source(s) to deposition sites, playing a key role in noble metal mineralization.

In hydrothermal ore systems, metal transport has traditionally been attributed solely to aqueous complexation (e.g., $Cl^-$, $HS^-$, $OH^-$)[1]. However, updated empirical thermodynamic and experimental modeling suggest that metal-rich melts are likely present at hydrothermal conditions[2,3], and can significantly upgrade the noble metal (e.g., in Au and Ag) endowment of ore deposits[4–7]. In agreement, analyses of micro- and nano-textures in ore minerals from a broad suite of hydrothermal deposits worldwide—including orogenic gold, skarn, intrusion-related gold, volcanogenic massive sulfide, porphyry Cu, greisen and granitic cupola, epithermal, and sub-epithermal (Supplementary Data 1, and references therein)—indirectly support the role of metal-rich melts in ore genesis. The presence of these melts in hydrothermal systems has recently been confirmed by Sousa-Guimarães et al.[8] and Jian et al.[9], who reported the co-occurrence of micron-sized solidified polymetallic melts and fluid inclusions within gangue quartz. They postulated that these melts formed late, near or at the deposition site, due to the inability of

hydrothermal fluids to transport such "large", high-density melt droplets. However, these studies (and others previous) have overlooked the role of nano-sized melt droplets as mineralizing agents transportable by upwelling hydrothermal fluids. Consequently, the available research has raised lingering fundamental questions: How, when, and where do metallic melts influence metal transport in hydrothermal mineralizing systems instead of aqueous complexes, or coupled with them? Can these melts exsolve from a deep-seated source (e.g., crystallizing magma) and be physically transported by hydrothermal fluids, or do they only form near or at the deposition site?

In this study, we aim to address these questions by analyzing polymineral inclusions—corresponding to solidified sulfide-sulfosalt melts—in the nanoscale realm, targeting solid micro and nanoparticles coexisting with fluid inclusion assemblages hosted by quartz from the El Hilo Au-Ag bonanza (Fig. 1a). El Hilo is part of the Natividad epithermal district in southern Mexico (longitude: 96.428°W, latitude:

[1]Instituto de Geología, Universidad Nacional Autónoma de México (UNAM). Ciudad Universitaria, 04510 Coyoacán, CDMX, Mexico. [2]Laboratorio Nacional de Geoquímica y Mineralogía (LANGEM), UNAM. Ciudad Universitaria, 04510 Coyoacán, CDMX, Mexico. [3]Instituto Andaluz de Ciencias de la Tierra (IACT-CSIC), Avda. de las Palmeras 4, 18100 Armilla, Granada, Spain. [4]Instituto de Geociencias, UNAM. Blvd. Juriquilla 3001, 76230 Juriquilla, Qro, Mexico. ✉e-mail: nacanoh@geologia.unam.mx

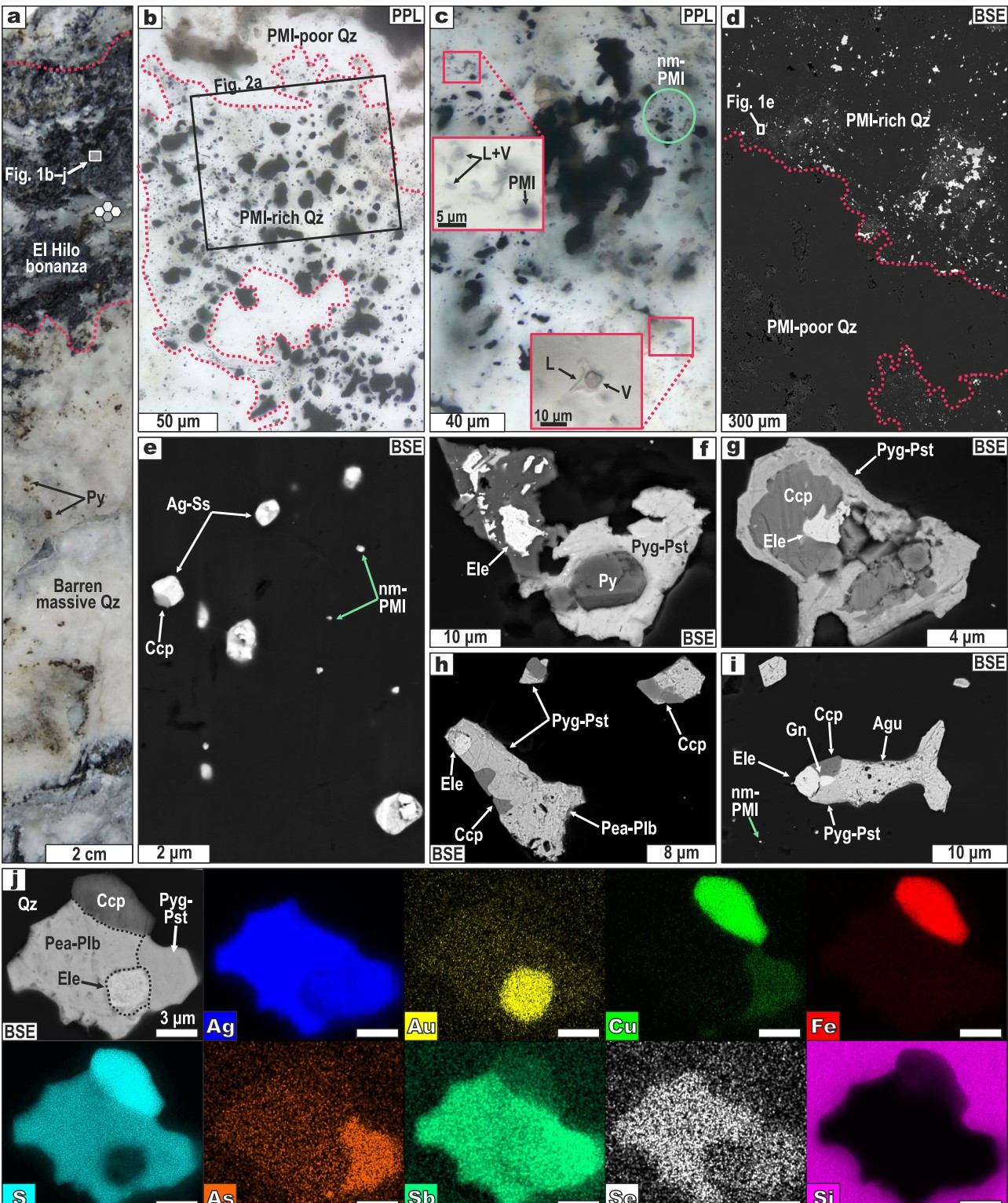

**Fig. 1 | Micron-sized polymineral inclusions (PMI) hosted by quartz in the El Hilo bonanza. a** Milky quartz vein that hosts El Hilo. **b** PMI-rich quartz in contact with PMI-poor quartz. Image enhanced by focus stacking. **c** PMIs along with liquid + vapor fluid inclusions, in the same fluid inclusion assemblage. **d**–**i** BSE images of nano-to-microPMIs. **j** Compositional map of one microPMI. Image type: BSE backscattered electron, PPL plane-polarized light, Mineral abbreviations: Ag-Ss Ag-sulfosalts, Agu aguilarite, Ccp chalcopyrite, Ele electrum, Gn galena, Pea-Plb pearceite-polybasite, Py pyrite, Pyg-Pst pyrargyrite-proustite, Qz quartz.

17.303°N), where mineralized veins are enclosed in Miocene porphyritic dacites and Paleozoic carbonaceous metasedimentary rocks[10]. This ~5-cm-wide Au-Ag bonanza ore zone grades up to 2 wt% Au and 31 wt% Ag, occurs within barren quartz, and lacks any internal lamination (Fig. 1a). Our approach reveals that nanomaterials (e.g., atomic clusters, nanomelts, nanoparticles) may be the most pristine evidence of a relationship between chemical and physical transport of noble metals in hydrothermal solutions. In addition, we provide numerical modeling evidence for the mechanical transport of metal-rich nanomelt droplets by upwelling hydrothermal fluids, thus explaining noble

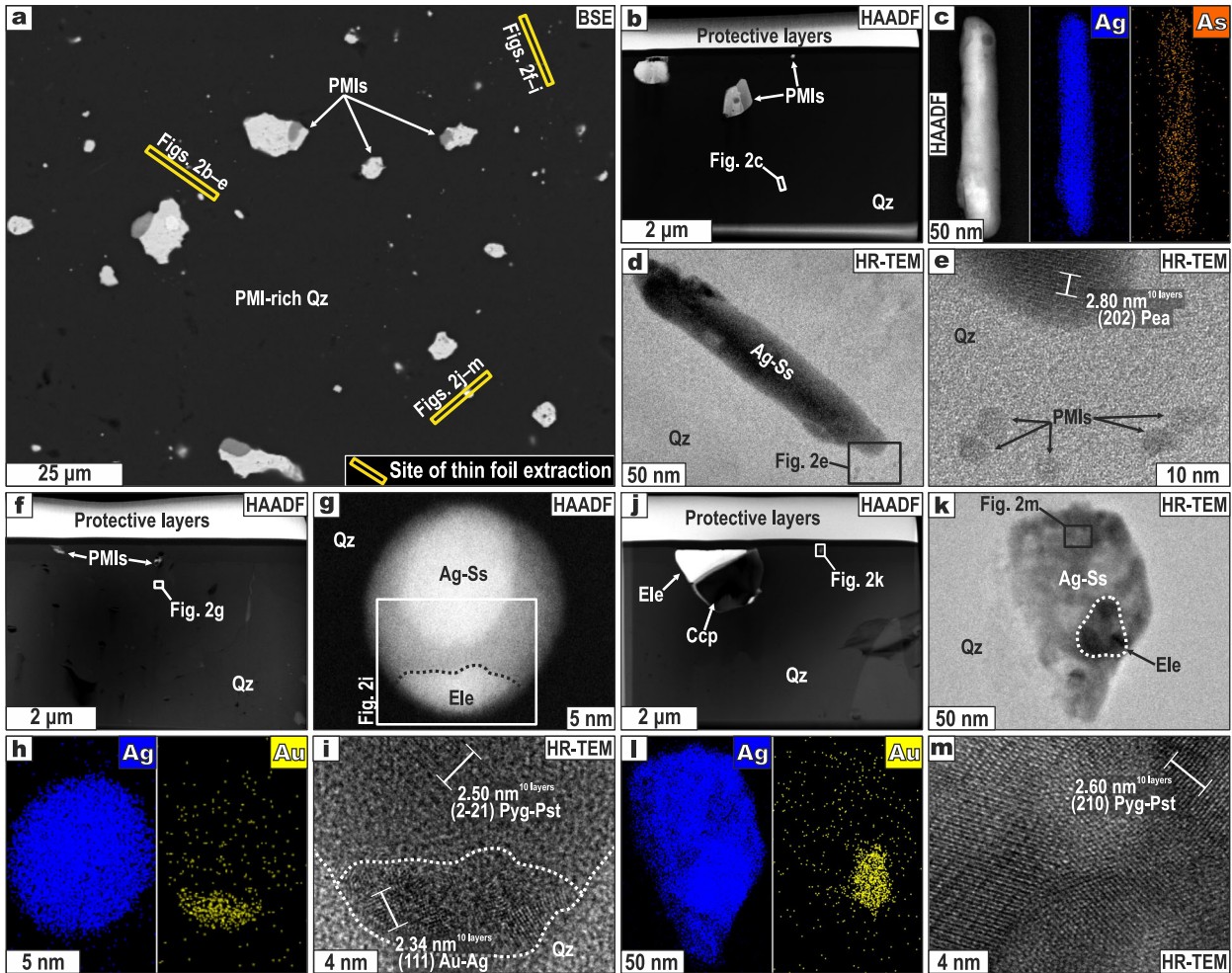

**Fig. 2 | Nano-sized polymineral inclusions (PMI) hosted by quartz in the El Hilo bonanza. a** BSE image showing the location of the studied thin foils. **b–m** Thin foils of quartz containing nano-scale PMIs that consist of sulfides, Ag-sulfosalts, and Au-Ag alloys (electrum). Measured *d*-spacings are superimposed on **e, i, m.** Scanning TEM elemental maps are shown in **c, h,** and **l.** Image type: BSE back-scattered electron, HAADF high-angle annular dark field, HR-TEM high-resolution transmission electron microscopy, Mineral abbreviations: Ag-Ss Ag-sulfosalts, Ccp chalcopyrite, Ele electrum, Pea pearceite, Pyg Pst pyrargyrite-proustite, Qz quartz.

metal enrichments upon ore deposition in a range of geological environments.

## Results and discussion

### Polymineral nanoinclusions track physical metal transport

Gangue quartz at the El Hilo bonanza hosts abundant nano-to-micron sized polymineral inclusions (PMI) of Ag(-Cu)-Sb-As-Se sulfosalts, electrum (Au-Ag), and Ag-Cu-Fe-Pb sulfides (Figs. 1b–j, 2a–m, Supplementary Fig. 1–5). This PMI-rich (mineralized) quartz crosscuts and replaces PMI-poor (barren) quartz along very sinuous interfaces and embayments, in some cases leaving remnants or "islands" of PMI-poor quartz surrounded by PMI-rich quartz (Fig. 1b–d, Supplementary Fig. 1). These textures suggest coupled dissolution-reprecipitation reactions between barren quartz and PMI-bearing hydrothermal fluids[11]. Field-emission scanning electron microscopic (FE-SEM) and transmission electron microscopy (TEM) inspection of PMI-rich quartz shows that PMIs ranging from ~5 nm (Fig. 2e, g, k) to ~40 μm (Fig. 1f–j) in diameter are randomly distributed through the grains. These PMIs comprise one to five minerals (Figs. 1e–j, 2g–l), including electrum, pearceite-polybasite [$(Ag_9CuS_4)(Ag,Cu)_6(As,Sb)_2S_7$], pyrargyrite-proustite [$Ag_3(Sb,As)S_3$], tetrahedrite-group minerals [$(Cu,Ag)_6[Cu_4(Fe,Zn,Cu)_2](Sb,As)_4S_{13}$], aguilarite ($Ag_4SeS$), acanthite ($Ag_2S$), chalcopyrite ($CuFeS_2$), galena ($PbS$), and pyrite ($FeS_2$).

Embayed electrum grains occur intergrown with sulfides or sulfosalts within PMIs (Fig. 1f–j), thus indicating that electrum crystallized early and was eventually resorbed during sulfide and sulfosalt crystallization. Chalcopyrite exhibits cuspate contacts with Ag±Cu-sulfosalts while galena occurs as islands within these sulfosalts (Fig. 1g–h, Supplementary Fig. 2), suggesting that sulfosalts postdated sulfides. Based on the analysis of 100 PMIs, we computed average compositions (± 1σ) of 61.3 ± 17.1 wt% Ag, 2.4 ± 7.7 wt% Au, 4.6 ± 3.6 wt% Sb, 2.3 ± 1.3 wt% As, 15.5 ± 4.9 wt% S, 1.8 ± 1.9 wt% Se, 6.0 ± 6.5 wt% Cu, and 3.9 ± 6.3 wt% Fe (Supplementary Figs. 2–3; Supplementary Data 2).

Nano-PMIs are bleb-like, some are perfectly rounded, while most micro-PMIs are irregular. Regardless of the size, minerals within PMIs show curvilinear or cuspate boundaries, usually joined at triple points (Figs. 1e–j, 2b–m, Supplementary Figs. 2–5). Semi-quantitative chemical composition obtained by SEM- (Fig. 1j, Supplementary Fig. 3) and scanning transmission electron microscopy (STEM)-electron dispersive spectroscopy (EDS; Supplementary Fig. 5) analyses indicates that PMIs are in the Ag-Au-Cu-Pb(-Fe-Zn)-As-Sb-S-Se multi-component system (Supplementary Data 2). Similar microtextural and chemical features of PMIs have been explained by the sudden deposition of dissolved metals and S from solution[12] or from fluid-mediated metal-rich melts[9,13,14]. Direct precipitation of focalized Au-Ag ores locally grading in the wt% scale as observed at El Hilo should have required

episodic metal deposition from large volumes of hydrothermal fluid. However, this is inconsistent with (1) the absence of laminated textures at El Hilo (Fig. 1a), which would account for episodic fluid injections, and (2) an extent of quartz veining and alteration haloes at El Hilo that is comparable to those of neighboring low-grade or unmineralized veins[10]. Alternatively, precipitation of those amounts of Au and Ag from a single fluid injection would have required unrealistically high precious-metal solubilities—which, for instance, approach ~0.1 ppm Au and 50 ppm $Ag_2S$ in fluids at 10 wt% NaCl equiv. and 350 °C[15]. Rather, our observations are more consistent with deposition from fluid-mediated melts enriched in metals, which is an effective mechanism to produce focused Au-Ag over-enrichments in orogenic gold deposits[6,9,16], even from fluids with minimal contents of precious metals (e.g., 0.2 ppb Au[5]).

To date, no experimental investigation fully elucidates the intricate relationships among cotectics, eutectics, and thermal boundaries within the multi-component system Ag-Au-Cu-Pb(-Fe-Zn)-As-Sb-S-Se, especially at the nano-realm. Nevertheless, the existence of metal-rich As-Sb-S-Se melts (referred to hereafter simply as "metal-rich melts", since metals account for >70 wt%; Supplementary Data 2) is consistent with fluid temperatures of ~270–400 °C[17] exceeding binary and ternary eutectic points of the simpler systems: Au-Pb (212 °C), As-S (310 °C), Au-Sb (360 °C), Ag-As-S (280 °C), Pb-Sb-S (240 °C), and Pb-As-S (305 °C)[2,4,18]. Accordingly, the stability of metal-rich melts in low-temperature (<400 °C) hydrothermal fluids is favored in (1) complex multicomponent systems containing low-melting-point chalcophile elements (LMCE), namely As, Sb, Se, Pb, Bi, etc.[2,5]; and (2) some metallic nanomaterials (e.g., Au-Ag alloy nanoparticles <10 nm in size), whose melting temperature decreases dramatically at the nanoscale compared with bulk counterparts[19,20]. Nanomaterials are considered by some to be the precursors of nucleation, growth, and, ultimately, mineral formation[21,22]. Therefore, nano- and micro-PMIs (Fig. 2b–m, Supplementary Fig. 5) represent solidified metal-rich melts that seem to have mediated ore mineral nucleation and growth during the formation of the El Hilo bonanza zone.

**Numerical testing of nano-micromelts as metal physical carriers**
Gangue quartz hosts abundant nano-to-micron scale PMIs (i.e., former metal-rich melts) along with fluid inclusions (Figs. 1c, 3a) with homogenization temperatures of 273–397 °C and salinities of 14–19 wt% NaCl equiv. (Supplementary Data 3)[17]. Frequently, PMIs co-occur with fluid inclusions, thus suggesting that melt precursors were originally entrained alongside their carrying aqueous fluids. In other words, these are evidence for heterogeneous trapping of mineralizing fluids carrying materials in suspension[23], which we interpret as complementary information that cannot be extracted from fluid inclusions alone, and thus their study is vital to understanding part of the evolution of ore-bearing fluids.

Upwelling hydrothermal fluids are able to mechanically transport materials (e.g., melt droplets or solid particles) only if the fluid-material interaction force balances the weight of the material, which is obtained for fluid velocities greater than a minimum fluidization velocity[24]. Minimum fluidization velocity ($v_f$; dashed lines in Fig. 3b) for a "bed of suspended particles"—in our case, a fluid containing melt droplets (see next section)—is given by[25,26]:

$$\Delta \rho g = \frac{150(1-\phi)\eta}{D_p^2 \phi^2} v_f + \frac{1.75\rho_f}{D_p \phi} v_f^2 \tag{1}$$

where $\Delta \rho g$ is the density difference between the material (i.e., metal-rich droplet ~6,300 kg/m³) and the fluid (837 kg/m³), $g$ is the acceleration due to gravity (9.81 m/s²), $\eta$ is the fluid viscosity (Pa s), $D_p$ is the particle diameter (m), and $\phi$ is the porosity—proportion of fluid relative to total volume (melt droplets + fluid). Figure 3b shows that melt droplets smaller than a few micrometers could have been

fluidized by fluids with ascent rates <10⁻² m/s at any given porosity. In addition, for a single melt droplet suspended in a fluid to be transported upward against gravity, upward flow rates $v$ (volumetric flow rate divided by total cross-sectional area) must also exceed the terminal settling velocity[25] ($v_t$; solid lines in Fig. 3b), as of

$$v_t = \sqrt{\frac{4}{3} \frac{D_p}{\rho_f C_D} g \Delta \rho} \tag{2}$$

where $C_D$ is a drag coefficient that depends on the Reynolds number:

$$Re = \frac{v D_p \rho_f}{\eta} \tag{3}$$

Fluid upwelling velocities on the order of 10⁻² to 10⁰ m/s have been calculated in some hydrothermal systems[27–29]. Taking ~10⁻¹ m/s as a conservative value, our model shows that upwelling hydrothermal fluids at different fluid regimes (laminar and transitional; $Re$ of 1, 10, and 100) would have fluidized and carried suspended metal-rich melt droplets in excess of 100 µm (Fig. 3b). This droplet size is larger than most PMIs trapped in quartz (Figs. 1–2), which average 4.5 ± 12.8 µm (2σ, n = 313; Fig. 3a), indicating that ore-bearing fluids at El Hilo would likely have been able to physically transport metal-rich melt droplets.

Flow rates higher than the fluidization velocity can be sustained, and hence the droplets remain in a fluidized condition, if the pressure loss in the hydrothermal system—due to interaction between any solids in suspension as well as friction losses due to complexities in the fracture network—is less than the initial pressure that started the flow[26]. Pressure loss $\Delta p$ along some vertical distance $L$ can be obtained from the Ergun equation (Eqs. 20-42 in Sissom and Pitts[25]):

$$\frac{\Delta p}{L} = \frac{\rho_f v^2}{D_p} \frac{1-\phi}{\phi^3} \left(150 \frac{1-\phi}{Re} + 1.75\right) \tag{4}$$

Setting $v = v_f$ and $D_p = 5$ µm (average PMI size; Fig. 3a) in Eq. 4, we computed the pressure changes across a fluidized bed of height $L$—here regarded as the distance to the source of melt droplets—at a given porosity (Fig. 3c). Although these calculations do not consider all the complexities inherent to these hydrothermal systems (e.g., variations in fracture aperture and rugosity), our results provide a first estimate of the pressure difference required to sustain the minimum fluidization condition over a given length. Vein-formation depths of ~1.1 km (30 MPa) calculated from fluid inclusions[10] added to a distance to the causative intrusion <4 km (i.e., <110 MPa)—arguable for intermediate-sulfidation epithermal deposits with no genetic link with high-sulfidation epithermal deposits[30,31]—yields a pressure gradient of ~80 MPa. This calculated pressure gradient exceeds the minimum fluidization condition for particles in high-porosity (>0.7) media (Fig. 3c), as expected for a fluid-dominated flow carrying melt droplets in suspension. Therefore, these observations suggest that minimum fluidization conditions could have been sustained over long distances (e.g., 4 km) from a given melt source to the deposition site (El Hilo).

During transit, metal-rich melts might have reacted with the fluid phase, resulting in either melt partial (or complete) dissolution by the fluids[32] or Au(-Ag) collection by the melts[5,6]. As already noted, the absence of laminated textures suggests that El Hilo may record a single fluid injection that was over-enriched (i.e., up to several wt%) in precious metals, both in solution and suspension. Such an over-enrichment implies that the fluids had met their solubility limits for precious metals—Au and Ag have low solubilities in hydrothermal solutions (on the order of ppb[1,33–35])—, thus avoiding the dissolution of metal-rich melts. Instead, PMIs with Ag contents ~61.3 wt% and Au ~2.4 wt% (Supplementary Data 2) may reflect that these melts probably collected noble metals from the hydrothermal solution. This agrees

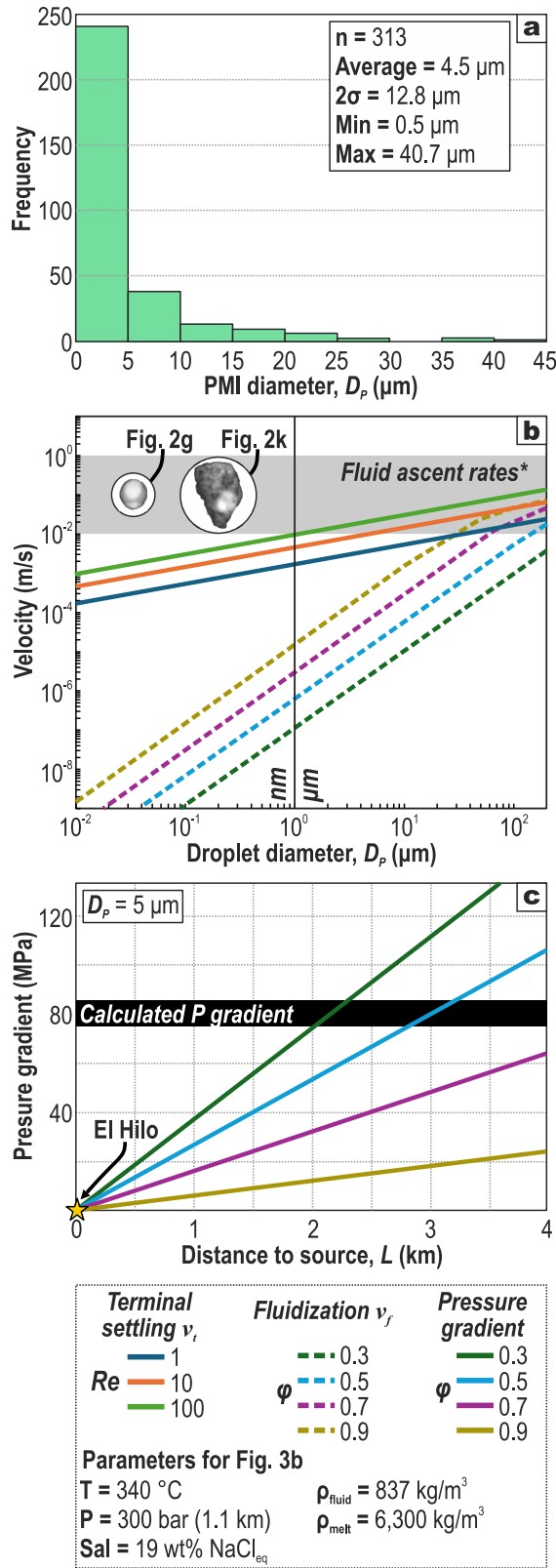

**Fig. 3 | Diameter of the studied PMIs and calculation of droplet settling velocity, fluidization velocity, and minimum pressure gradient for fluidization.**
**a** PMI Feret's diameter distribution for 313 PMIs (Supplementary Data 2). **b** Droplet diameter vs. velocity plot based on Eqs. 1–3 for fluidization and settling velocity of PMIs (i.e., solidified nano and micromelts; Supplementary Data 3). Fluidization velocity was calculated for a variably porous ($\varphi$ = 0.3, 0.5, 0.7, and 0.9) "bed" of melt droplets (voids filled with hydrothermal fluid), and settling velocity was calculated for different fluid regimes ($Re$ = 1, 10, and 100). See examples of nano-sized PMIs in insets. *Fluid ascent rates according to Okamoto and Tsuchiya[29], Dissanayake et al.[27], and Delaney et al.[28]. **c** Distance vs. pressure gradient diagram showing minimum fluidization conditions for the "bed" of droplets at different porosities.

Once on site, boiling and conductive heat loss might have destabilized metals in solution while promoting metals partitioning into the suspended melt droplets[5]. Fluid turbulence due to boiling might have increased the Brownian motion of suspended (nano)melts, thus enhancing their interaction with the fluid (i.e., metal scavenging) as well as their coalescence—similar to metal deposition from colloidal suspensions[17,36,37]. Consequently, the concurrent deposition of suspended metal-rich melts and their subsequent solidification to PMIs as the temperature decreased, likely upgraded the "regular" mineralization process through precipitation from brines, thus leading to the Au-Ag bonanza.

**Provenance of metal-rich sulfide-sulfosalt (nano)melts**
El Hilo contains metal-rich nano and micromaterials (i.e., PMIs, former melts) that were probably transported within hydrothermal fluids from a certain source(s) to the deposition site (Fig. 3a–c). Similar to other intermediate-sulfidation epithermal deposits[31], the causative magmatic body for the Natividad deposit is not exposed and is thus interpreted as distally located[10,17]. In this rationale, nano-melt droplets could have originated near the magmatic source and/or once the fluids reached the epithermal environment, before arriving at the deposition site (Fig. 4a). In fact, metal-rich droplets could have originated even deeper, in the middle–lower crust or mantle[38,39], and then transported to the uppermost crust by silicate melts[40].

A magmatic source for metal-rich (nano)melts is supported by sulfides and sulfosalts from El Hilo and neighboring veins showing $\delta^{34}S_{VCDT}$ between −3.2‰ and −0.3‰[10], within the −3‰ to 3‰ range of mantellic S[41]. Fluid inclusion data indicate that these melts were trapped from fluids with temperatures of 273–397 °C and salinities of 14–19 wt% NaCl equiv[17]. In the absence of evaporites, epithermal fluids at these temperatures and salinities represent mixtures of magmatic metal-rich brines and low-salinity fluids (e.g., deeply evolved meteoric waters or late condensed magmatic vapors)[31,42,43]. These magmatic brines are characterized by high temperatures (up to 650 °C), salinities (26–75 wt% NaCl equiv.), and densities (>1.3 g/cm³)[15]. Hence, magmatic brines typically stall above crystallizing intrusions after their release[44] (Fig. 4a, inset 1), which may induce rock fracturing due to the overpressure[26,45] (dotted red line in Fig. 4b). The magmatic fluid release takes place near the brittle-ductile threshold (~400 °C), out of reach of external fluids, and maybe parental to potassic cores of porphyry-type mineralization[15,44–46]. Thus, mineral assemblages of potassic cores in porphyry deposits could be regarded as a pristine archive of hydrothermal processes (e.g., fluid release) that take place in a purely magmatic environment. Nanomelt droplets of Au-Ag(-Pd-Pt)-Te-Bi-Se-S have been reported in potassic cores at Elatsite[47] (Bulgaria) and Skouries[48] (Greece) porphyry Cu(-Au) deposits (Supplementary Data 1). Whether the El Hilo bonanza is associated with underlying porphyry-type mineralization, this suggests that metal-rich nanomaterials can indeed be produced during magmatic fluid release from fractionating magmas[49,50], and then be stored at depth in hyper-saline magmatic fluids (Fig. 4a, inset 1). Consistently, recent numerical simulations[51] and textural observations[32] have shown that Cu-Fe-rich

with the preferential partitioning of noble metals from the aqueous solution to the melt, which can be dramatic in some chemical systems (e.g., $D_{Au}^{liquid\,Bi/aqueous\,fluid}$ ~ $4 \times 10^8$)[3,5]. Further, nano-scale PMIs (Fig. 2b–m, Supplementary Fig. 5) consist of the same ore minerals as micron-sized PMIs (Fig. 1e–j, Supplementary Fig. 2), thus suggesting that nanomelt (~5 nm) droplets may have coalesced to form larger ones (~40 μm) while migrating or upon deposition.

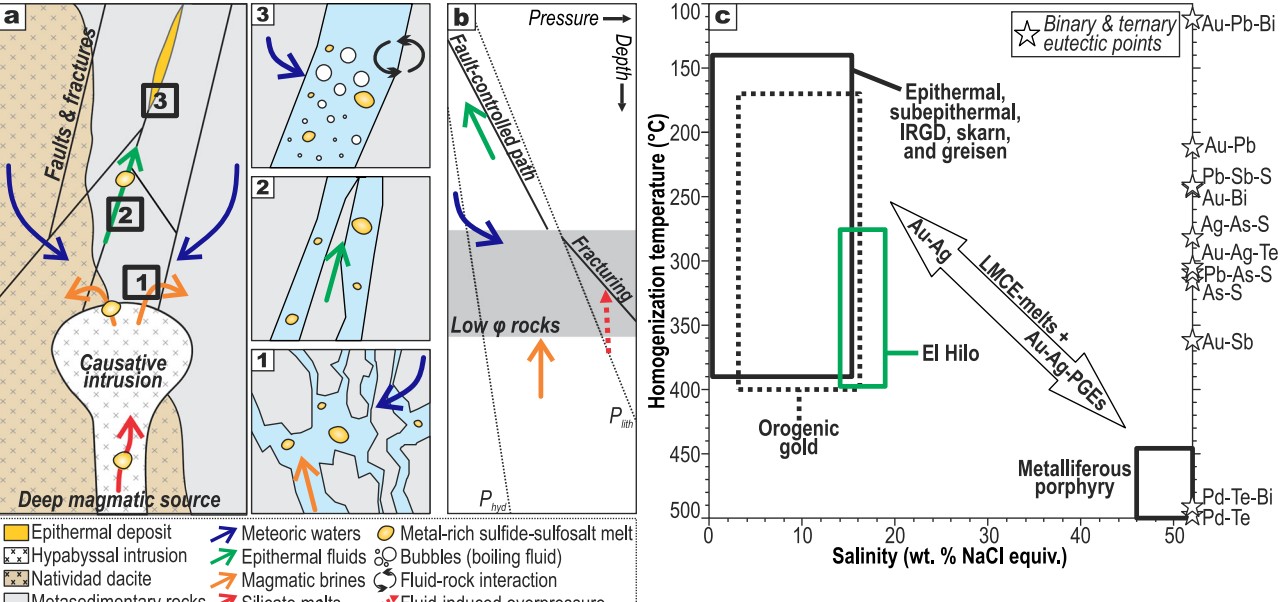

**Fig. 4 | Occurrence and formation of metal-rich sulfide-sulfosalt melts in hydrothermal ore deposits. a** Illustrative model showing the possible provenance of metal-rich melts from the El Hilo bonanza, with close-ups in insets to (1) deep-seated magmatic brine reservoir, (2) hydrothermal fluids rising through the fracture network, and (3) shallow fluid boiling, buffering due to fluid-rock interactions, and mixing with meteoric waters. **b** Schematic depth vs. pressure diagram showing inferred processes involved in fluid (+ melt) migration. Based on Oliver et al.[26] and

Sibson[64] **c** Salinity vs. homogenization temperature plot showing fluid inclusion data from several types of deposits wherein metallic melts contributed to noble metal mineralization (see details in Supplementary Data 1). Eutectic points for LMCE-noble metal binary and ternary systems are included[2,4,5,9,18,65]. IRGD: intrusion-related gold deposit, LMCE: low-melting-point chalcophile elements, PGE: Pt-group elements (e.g., Pt, Pd, Rh).

(+Pd+Au) sulfide droplets—and other dense metallic materials[52]—can float alongside vapor bubbles within andesitic–dacitic melts, and then transfer metals to magmatic brines. Over time, isotherms migrate downward as crystallization of the magma body proceeds[53] and magmatic brines can mix with low-salinity fluids, thus resulting in a low–moderate-salinity and metal-rich fluid that migrates along fracture networks (Fig. 4a, b) and feeds overlying epithermal systems (Fig. 4a, inset 2)[31,42,43]; this was probably the case for the Natividad epithermal deposit[10]. In agreement with our model (Fig. 3b–c), this suggests that if metal-rich nanomelts were present in magmatic brines at depth (e.g., 4 km), they could have been fluidized by upwelling low–moderate-salinity fluids and subsequently well up to the epithermal environment.

Metal-rich magmatic-hydrothermal fluids must transit fracture-controlled pathways while rising through the epithermal environment, wherein fluids are prone to boiling, fluid buffering by interaction with wall rocks, and mixing with external waters[31,42,45]. At El Hilo, Au-Ag ores are associated with the occurrence of (1) coexisting fluid-rich and vapor-rich fluid inclusions, which indicates boiling; (2) methane in some fluid inclusion assemblages, suggesting the assimilation of carbonaceous materials; and (3) fluids with $\delta^{18}O$ from 1.7‰ to 4.4‰ and $\delta^{13}C$ from −6.0‰ to −4.4‰, showing mixtures of magmatic fluids and meteoric waters[10,17]. Some authors have found that $H_2S$ (plus other volatiles) loss due to boiling[1,31,42] and fluid buffering by carbonaceous metasedimentites[54] are key to concentrating Au(-Ag) in hydrothermal deposits. Thus, fluid inclusion and C–O isotopic data from El Hilo suggest that boiling, fluid buffering, and/or mixing could have produced precious metal-rich nanomelt droplets in the epithermal setting (Fig. 4a, inset 3). Consistently, authors have found that these processes promote the formation of nano-sized metallic materials in hydrothermal fluids[12,36,50], which may constitute colloidal suspensions that can be transported within the fluids and generate bonanzas[49,55,56]. Indeed, our numerical models (Fig. 3b, c) support this mechanism of mechanical transport to explain the formation of the El Hilo bonanza.

Nevertheless, this scenario does not exclude the further formation of nanomelts upon precipitation from solution, given that some Au and Ag were probably transported (alongside melts) by complexing ligands (e.g., $AuHS_2^-$ and $AgCl_2^-$) that are readily destabilized by boiling and conductive heat loss, among other phenomena[15].

In summary, our geological, fluid inclusion, mineralogical, and numerical modeling results suggest that metal-rich sulfide-sulfosalt melt droplets were produced at a certain distance from their ultimate deposition site, and were then transported in suspension by upwelling hydrothermal fluids. However, it remains unclear how far melt droplets were produced (i.e., near causative intrusion vs. along the way). Figure 4b summarizes examples of hydrothermal ore deposits where metal-rich melts contributed to Au ± Ag mineralization (Supplementary Data 1, and references therein). Au-Ag-LMCE(-S) melts are commonly associated with ore-forming fluids with low–moderate temperatures (~100–400 °C) and salinities (<20 wt% NaCl equiv.), which are parental to various types of magmatic-hydrothermal ore deposits[15,57,58] (Fig. 4b). These deposits are produced by mixtures of magmatic metal-rich brines and low-salinity fluids and may potentially undergo comparable evolutionary processes to those described for El Hilo[31,42,43,57,59] (Fig. 4a). We thus postulate that these processes represent a complementary mechanism of metal transport (alongside metals in solution) to explain the enrichment of noble metals in hydrothermal fluids, playing a key role in the formation of high-grade mineralization across diverse geological environments.

## Methods

### Scanning electron microscopy and focused-ion beam
Polymineral inclusions (PMI) hosted by quartz were first identified with a petrographic microscope, before using electron microscopy. SEM analyses were performed with a field emission QemScan 650 (Thermo Fisher-FEI) SEM and a JSM-IT800 super hybrid lens version (JEOL) SEM, both at the Centro de Instrumentación Científica (Universidad de Granada), Spain. SEM images are backscattered electron images

acquired with solid-state detectors installed in both SEMs. The instruments are equipped with energy dispersive spectrometer (EDS) silicon drift detectors. Accelerating voltage and beam current were optimized depending on the size of PMIs to ensure high-quality images and an adequate number of counts for EDS measurements.

After selecting the best areas for thin-foil extraction, we used an FIB-SEM to prepare three thin foils of quartz containing nano- and micron-scale PMIs (Supplementary Fig. 4). FIB-SEM analyses were carried out with a Dual Beam FEI Thermo-Fisher Scientific, model Helios 650 at the Laboratorio de Microscopías Avanzadas (LMA; Instituto de Nanociencia de Aragón–University of Zaragoza), Spain. The selected regions of interest were covered with a first layer of C (~300 nm) and a second layer of Pt (~1 μm), which functioned as protection during the milling, polishing, and extraction of the thin foil. The bulk material was first removed on both sides of the lamellae with a Ga + ion milling at 30 kV and 2.5 nA current, and the subsequent polishing with a 30 kV current at 0.23 nA. The final polishing step was completed by milling the thin foils with a 5 kV current at 68 pA. The electron transparency was monitored by an Everhart-Thornley SE detector and using a 5 kV electron beam. After achieving the electron transparency, the thin foils were rapidly polished using a low energy 5 kV current at 10 pA to reduce the amorphization until a final thickness of ~80 nm was attained. Subsequently, the thin foils were undercut with a 30 kV at 2.5 nA current, lifted out, and transferred from the sample to a TEM Cu grid using an OmniProbe nanomanipulator with a tungsten tip. To weld the thin foils to the tungsten tip and the TEM grid, an ion-beam-assisted Pt deposition was performed.

### High-resolution transmission electron microscopy
A probe-corrected Titan (Thermo-Fisher-FEI) TEM equipped with a XFEG field emission gun was used to analyze the thin foils at the LMA (Zaragoza, Spain). This microscope is equipped with a high-brightness X-FEG and a spherical aberration Cs-corrector at the condenser system (probe-corrected). Areas of interest within the thin foils were imaged using (1) high-angle annular dark-field (HAADF) to obtain Z high-contrast images by scanning transmission electron microscopy (STEM), and (2) HRTEM images to describe textural features of PMIs and to index minerals. The Titan was running at 300 kV while HRTEM images were acquired using the Gatan CCD Camera. The composition of PMIs was obtained with EDS analyses using an Ultim Max detector (Oxford Instruments). All these data were treated using the AZtecTEM software package (Oxford Instruments).

### Estimation of polymineral inclusion composition
The composition of PMIs was calculated for 100 PMIs ranging in area between 0.5 and 103.2 μm². Minerals within PMIs were first identified using EDS spectra. Then, we estimated the areal proportion of each mineral within each PMI using the ImageJ-based Fiji software[60]. Finally, we computed the chemistry of every PMI using electron probe micro-analyzer data for electrum, pearceite-polybasite, pyrargyrite-proustite, tetrahedrite-group minerals, and aguilarite from Cano et al.[10], as well as ideal compositions for chalcopyrite, galena, and pyrite. All these calculations are presented in Supplementary Data 2.

### Fluidization velocity, settling velocity, and pressure gradients
We estimated the fluidization velocity using Eq. 1, settling velocity using Eqs. 2 and 3, and pressure gradients to maintain minimum fluidization conditions using Eq. 4. These equations contemplate several variables, including (1) fluid and PMI density difference ($\Delta\rho$), (2) fluid viscosity ($\eta$) and density ($\rho$) at a given temperature, pressure, and salinity; (3) the gravitational acceleration ($g$), (4) the particle diameter ($D_p$), (5) the porosity ($\varphi$), (6) the drag coefficient ($C_D$), and (7) the Reynolds number ($Re$). (1) A fluid density of 837 kg/m³ was estimated using fluid inclusion data from Cano et al.[10,17] for the El Hilo bonanza zone in the HokieFlincs_H2O-NaCl software[61] (Supplementary Data 3).

A PMI density of 6300 kg/m³ was computed using mineral proportions for each PMI combined with mineral densities from Mindat.org[62] (Supplementary Data 2). This density represents a maximum estimation, considering that molten PMIs (i.e., metal-rich sulfide-sulfosalt melts) should have a slightly lower density. (2) Fluid viscosity was computed using experimental data by Kestin et al.[63] for H₂O-NaCl solutions at variable temperature, pressure, and salinity. We calculated fluid viscosity at 0 (0 mol/kg), 10 (2 mol/kg), and 19 (4 mol/kg) wt% NaCl considering an average temperature of 340 °C and pressure of 300 bar (~1.1 km), as indicated by fluid inclusion data from Cano et al.[10,17] (Supplementary Data 3). (3) The gravitational acceleration was set at 9.81 m/s². (4) The particle diameter was estimated for 313 PMIs from Assemblages 1 to 8 (Supplementary Fig. 2) by applying the Thresholding option of the Fiji software, which was possible due to the large difference in brightness between PMIs and the hosting quartz in back-scattered electron images. This diameter corresponds to the Feret's diameter, which is the longest distance between any two points in the PMI. (5) The porosity was set at 0.3, 0.5, 0.7, and 0.9 to reflect variable fluid ($\varphi$)/melt droplets + particles ($1-\varphi$) ratios within the fractures, in order to provide a more comprehensive representation of the hydrothermal system; (6) the drag coefficient was calculated according to Sissom and Pitts[25] for spherical particles (Fig. 20–2); non-spherical particles (cylinders, prisms, disks, etc.) would lead to smaller settling velocities ($v_t$)[25]. (7) The Reynolds number was set at 1, 10, and 100 to comprise laminar and transitional flows. See Supplementary Data 3 for more information on calculations.

## Data availability
The data generated in this study are provided in the article and Supplementary Materials, which are also available at https://doi.org/10.6084/m9.figshare.28451141.

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

## Acknowledgements

This work benefited from grant NANOMET PID2022-138768OB-I00 funded by MCIN/AEI/10.13039/50110001133 and "ERDF A way of making Europe" by the "European Union" (J.M.G.J.), and from grant IN102123 funded by PAPIIT-DGAPA-UNAM (A.C.). Additional funding to N.C. was provided by CONAHCyT (Ph.D. grant), the Posgrado en Ciencias de La Tierra-UNAM, and the Geological Society of America (grant 14012-24). Authors acknowledge the use of instrumentation as well as the technical advice provided by the National Facility ELECMI ICTS node Laboratorio de Microscopías Avanzadas at the University of Zaragoza. Isabel Sánchez (CIC-UGR), Laura Casado, Alfonso Ibarra, Rodrigo Fernández (LMA-Unizar), Juan González García (UCA), Edith Fuentes Guzmán, and Juan Tomás Vásquez (UNAM) are acknowledged for their assistance during sample preparation and laboratory analyses. Baltazar Chávez, Senén Benítez, and the team at Natividad (Oaxaca, México) are acknowledged for their assistance during the stay at the mine facilities.

## Author contributions

N.C., J.M.G.J., A.C., E.M.C., and E.G.P. conceived the study. N.C. collected, selected, and prepared the samples. N.C. and J.M.G.J. conducted laboratory analyses. E.M.C. and N.C. constructed the numerical models. N.C., J.M.G.J., A.C., E.M.C., and E.G.P. wrote the manuscript.

## Competing interests

The authors declare no competing interests.
