## [Peer Review File · Nature Communications]

Transient non-soluble noble metal transport in hydrothermal ore systems

Corresponding Author: Dr Néstor Cano

Version 0:

Reviewer comments:

Reviewer #1

(Remarks to the Author)

Manuscript: A transient non-soluble noble metal transport in hydrothermal ore systems

The manuscript presents an intriguing study on the transport of noble metals by polymetallic melts in hydrothermal ore systems, proposing this as a complementary mechanism to metal-bearing ligands in solution. The authors provide evidence for suspension-like transport of nano-to-micron sized polymetallic melts within epithermal fluids, leading to the formation of polymetallic inclusions in quartz. While the manuscript introduces exciting ideas, there are some areas that require further clarification and discussion.

Major Issues

Data Interpretation:

The authors interpret the polymetallic inclusions in quartz as indicative of deposition from a melt (line 110). Since this textural interpretation forms the core of their model, it is essential that it is robust and well-supported. It is possible that these textures could also result from the sudden deposition of dissolved metals and sulphur from hydrothermal fluids during a boiling event. To strengthen their interpretation, the authors should provide a more thorough discussion, demonstrating that no other deposition processes could account for the observed textures. Upon reviewing the provided figures (Figure 1.f, g, h, i, j, and Figure 2.a and b), the polymetallic inclusions do not appear particularly blebby. More evidence and detailed explanation are needed to substantiate the claim that these textures are indeed a result of deposition from a melt.

Composition Discrepancy:

The abstract states that nano-to-micron sized polymetallic melts within epithermal fluids form bleb-like polymetallic inclusions of Au-Ag(-As-Se-Sb) (line 24). However, the data presented in the manuscript indicate that the polymetallic inclusions are composed of "Ag(-Cu)-Sb-As-Se sulfosalts, electrum, and Ag-Cu-Fe-Pb sulfides." Furthermore, line 108 describes a "Ag-Au-Cu-Pb(-Fe-Zn)-As-Sb-S-Se multi-component system." The presence of sulphur is significant as it influences the behaviour of the polymetallic melt. This inconsistency between the abstract and the actual data presented needs to be addressed to maintain the reliability of the conclusions drawn.

Transport of PM Droplets:

The manuscript suggests that polymetallic (sulfosalt and sulfides) material is mechanically transported by rising hydrothermal fluids as melt droplets (lines 182-184, 191-193) from a "distally located" magmatic intrusion to the surface. While the authors demonstrate that velocity calculations make such transport possible, it is not discussed how such material would remain in a chemically stable suspension within the hydrothermal fluid. Existing experimental studies show that sulfide melts are not easily transported (Rose and Brenan, 2001). The manuscript does not sufficiently address the chemical processes by which the melt droplet could be kept in a stable suspension in the aqueous epithermal fluid ("LMCE-bearing metallic (nano)melts could be kept suspended," line 233). The potential for the melt to react with the aqueous fluid, rather than remaining inert and stable, needs more discussion.

Additionally, how the nano-to-micro size droplets would percolate through kilometres of rock without reacting with the wall rock or being mechanically stopped by limited porosity requires further elaboration.

To demonstrate the transport of the melt, the authors should explain how it can be in both mechanical and chemical

equilibrium within the fluid.

Specific Comments

Line 24: The statement about the composition of polymetallic inclusions does not match the data presented. The manuscript needs to clarify that the inclusions are primarily sulfide melts, which alters the interpretation of the findings.

Line 34: The references cited (Econ. Geol. 99, 1063–1084 (2004); Toothe 2008 and 2011; Frost 2002) are used to support the hypothesis of polymetallic melts in hydrothermal fluids. However, these studies primarily discuss local migration within host rocks or low-melting point chalcophile elements, not the long-distance transport in aqueous fluids proposed by the authors. The manuscript should critically assess these references and clarify their relevance to the current study.

Line 78: There is a critical need for a more detailed discussion on the phases referred to as PMI by the authors: Ag(-Cu)-Sb-As-Se sulfosalts, electrum, and Ag-Cu-Fe-Pb sulfides. The manuscript must differentiate between polymetallic melts and sulfide melts and address their transport mechanisms individually.

Line 105: More detailed descriptions of the textures are needed to convince the reader that the textures are indeed the remnant of melts.

Line 122: Can you be more specific here regarding what early segregation refers to? Do you mean early segregation from a parental magma?

Line 131: This statement is slightly confusing. It does not appear that your data shows sulfides PMI within a fluid inclusion. Rather, they seem to be occurring within the same quartz crystal.

Line 169-170: This sentence needs clarification. It is unclear what is interpreted to happen to metals in suspension and how it relates to polymetallic melt.

Line 191-193: This part needs more explanation. Are the polymetallic melts at that stage sulfide melts? (It should be, considering your data.) More discussion is needed to explain how the sulfide melts/droplets can be segregated from the magma into the aqueous fluid.

Line 232-233: There is a discrepancy between the literature cited and the interpretation made by the authors. The metal-rich brines referred to in line 232 are brines in which metals are dissolved as chloro-complexes and not suspended as nanomelts. There is no data to support the claim that nanomelts are present in the brines. A discussion is needed to address how such nanomelts would be stable in this environment and how they might have been extracted from the magmatic precursor.

Figure 1: The manuscript lacks a discussion on quartz dissolution textures presented in Figure 1, which could provide insights into the deposition mechanisms and the environment of formation.

Figure 4: The composition of the rising fluid and its ability to stabilise polymetallic inclusions during transport must be addressed. This includes discussing the role of fluid-rock interactions and the impact of wall rock porosity.

In conclusion, the ideas proposed are novel and exciting and could help explain ultra-high-grade enrichment, which remains to be understood. However, I believe that more work needs to be done on this manuscript to convincingly explain how the proposed transport is chemically and physically possible.

Reference:

Rose, L. A., and Brenan, J. M., 2001, Wetting properties of Fe-Ni-Co-Cu-OS melts against olivine: Implications for sulfide melt mobility: *Economic Geology*, v. 96, no. 1, p. 145-157.

Reviewer #2

(Remarks to the Author)

See attached review comments and edited PDF

Reviewer #3

(Remarks to the Author)

This manuscript introduced a new mechanism of enrichment of noble metals in magmatic-hydrothermal ore forming systems. The authors propose that nano-micro sized melts may be present and suspend in hydrothermal fluids and be transported by fluids to form mineralization. Evidence provided includes SEM, EDS, TEM and EPMA analyses, etc. The idea reads interesting with relevant data and observations. However, I have some concerns related to whether the melts could be stable in the fluids at the T-P-salinity conditions as discussed, as well as if the calculation of the fluids' capability to transport and maintain melts suspension is accurate enough. Please see below:

1. It is important to consider the chemical stability of the noble metal melts in the hydrothermal fluids with relatively high T and salinity. At conditions of El Hilo fluids, e.g., 100-400°C, up to 20wt% NaCl, the fluids may dissolve significant amounts of metals, in particular when considering that chloride complexes are thought to be important species for Au, Ag, and PGEs in Cl-dominated fluids and contribute to their hydrothermal transportation. Hence, it is important to provide constraints on if the nano-micro metal melts can be chemically stable as well in such fluids. I suggest the authors to provide thermodynamic calculations of the relevant metals and if possible, the fluid composition measurement of the fluid inclusions to demonstrate what the solubility of metals is.

2. The calculation of the capability of upwelling fluids to mechanically transport particles is based on the assumption of the fluids of a fixed density of 840kg/m³ at 340°C and 300 bar. This was then applied to calculate the settling velocity of fluids with various salinities. However, it is important to note that the salinity has a significant influence on the fluid density as well. I have done some calculations myself: for fluid at 340°C, 300bar, the density of fluids for 0m, 2m, and 4m NaCl would be

670, 793.7, and 879.14kg/m³ respectively. There is a big variation there, and what is the influence of such on the settling velocity? The authors state that the salinity has an insignificant impact on the settling velocity because they ignored salinity's impact on density. I suggest the authors to revise their calculations and provide a more accurate estimate. In summary, although the proposed idea is new and interesting, I suggest some of the key information described above is missing. These are essential to demonstrate that the metals can really be stable and transported by the upwelling fluids. This information should be provided to further evaluate the proposed model before it can be accepted.

Version 1:

Reviewer comments:

Reviewer #1

(Remarks to the Author)

I have reviewed the revised manuscript and am satisfied with the authors' responses to my comments. I believe the points raised during the initial review have been addressed thoroughly and with appropriate consideration. The revisions have significantly improved the clarity and strength of the manuscript.

I have no further comments and consider the manuscript, in its current form, to be suitable for publication.

Summary of other concerns from this revision, in light of Reviewer #3's comments:

Reviewer #3 highlights an important point about the need to provide the density of the PMs. This is critical since these metals are very dense, which affects their ability to be transported upwards. The authors could explore whether silica gel (as described in McNab et al, 2024 Coprecipitation of amorphous silica and gold nanoparticles contributes to gold hyperenrichment- GEOLOGY) might carry these metals, potentially mitigating the density issue. However, the gel would also need continuous porosity for transport.

It is unclear whether metals are transported as a melt suspended in fluid or as solid nano- or microparticles (example line 257). If transported as a melt, what processes cause solidification at the site of mineralisation? The explanation provided—boiling leads to deposition—is vague and there is confusion between the coalescence of melt vs solid particles. Both references cited here 16 and 35 are describing the coalescence of solid metallic nanomaterial not melts. It is likely that the coalescence mechanism between a solid particle and melt differ and this point should be addressed in the discussion.

There are some inconsistencies in the description of the material being transported in the proposed model. For instance, in lines 126 and 135, the authors refer to a "metal-rich melt," while later, in lines 246 and 263, they describe "sulfide melts" or "metal-rich sulfide nano melts." This distinction is significant and needs to be clarified.

In L244, the manuscript states that precious metals avoid dissolution in hydrothermal fluids, which is likely due to their low solubility but it is not the case for sulfur that has a high solubility in these systems. Then how does S remain in the melt phase rather than transitioning to the fluid phase? The stability of a sulfide melt under these conditions requires further explanation, as sulfide melts are typically associated with magmatic rather than aqueous environments.

The authors also reference the Bi collector model, where precious metals partition into a Bi melt phase in an aqueous system. While this is a well-documented process, the manuscript instead describes a sulfide melt, which may behave differently in such an environment. The stability of a sulfide melt under hydrothermal conditions, particularly its ability to persist and transport metals, needs to be addressed further in my opinion.

Additionally, if the melt is described as "metal-rich," does it imply a sulfide-dominated composition or one dominated by precious metals with low solubility? These two scenarios could have distinct implications for melt stability, transport properties, and deposition mechanisms. Clear distinctions between these terms and their respective roles in the proposed model would help resolve potential confusion and strengthen the study's argument.

(Remarks on code availability)

Reviewer #2

(Remarks to the Author)

I've reviewed the changes the authors have made since the previous review and I can see that they've put a lot of effort into addressing each comment. With regards to my previous comments, my main criticism was around the too-simplistic use of Stoke's Law as a way of exploring the carriage of micro- and nano-scale particles. The authors have completely redone the relevant part of the manuscript and it is now much more rigorous. As I read through this new material, I was wondering, if the particles are so easy to transport, what would make them accumulate? And then, at the end of the section (Page 14, lines 253-260), the authors discuss boiling at the site of accumulation and metal-upgrading. I guess this is a reasonable way to round out this section, and that question of nano-droplet/particle accumulation is one that the research community will continue to ponder for some time. I am happy to recommend that the article be published now.

(Remarks on code availability)

Reviewer #3

(Remarks to the Author)

The revised manuscript provides some good improvements from their previous version. More evidence was added to support the mechanical transport of PMIs. This is good, but I think there are still some more careful justifications are needed regarding this new transport mechanism of precious metals. The manuscript should not be published before these points can be well addressed.

These are:

1. The author in their reply and new calculations show that porosity has a more significant influence on fluidization and settling velocities. This is apparent. However, as the calculated porosity was 'set' to 0.3, 0.5, 0.7 in their calculation, no explanation was provided how such numbers are based on. Well, to me, even 0.3 looks like a big overestimate of porosity by looking at their SEM of PMIs. The electrum was even more dense and less porous compared to other parts of the PMIs. As the porosity of PMIs is showing such important influence on their mechanical transport in fluids (1 to >1.5 orders of magnitude by looking at their new figure, comparing PMI porosity 0.3 to 0.7), it is necessary to provide the data on the PMI density: what is the PMI porosity in your sample?
2. The question then comes to the density calculation of the PMIs. It is clear if the PMIs are too dense then they will precipitate fast, not possible at all for them to be transported by a fluid. The provided density calculations of PMIs were based on mineral data (solid) and the numbers were calculated by using mixtures of such through compositional data. There are two major flaws there: (1) how different will it be for a melt mixture and a solid mixture when they have the same composition? (2) the observed particle size seems to be solid rather than a polymetallic melt, which may have significantly smaller size but higher density. If such, it will be more difficult to transport them in fluid.
3. Then, since the electrum are more dense and has higher density than other parts of PMIs, the author should explain better how these things with such contrasted physical properties can be hold together then a fluid is trying to flush them. It is very likely the less dense part which also have higher porosity will be effectively separated with Ag-Au electrum.
4. The temperature range for these melts was assumed to be <400°C, yet, the melting point of mentioned phases (see their supplementary table S2) are mostly much higher (eg, Ag is ~960°C, Au is 1064°C). The author need to explain better in the text how such melt can be stable in such a low T circumstances.
5. The impact of the proposed mechanism should be better discussed. It seems that even if the mechanism is true, it is more of a localized phenomenon and may not really explain large-scale precious metal mineralization, especially for Au. It may explain El Hilo, but how to apply this to other cases? Please provide more insights.

(Remarks on code availability)

Version 2:

Reviewer comments:

Reviewer #1

(Remarks to the Author)

Thank you for your thorough revisions and for clarifying the terminology throughout the manuscript. The improved consistency in describing the transported material strengthens the study and makes the interpretation more accessible. I am overall satisfied with the revisions and am happy to recommend the manuscript for publication.

Reviewer #3

(Remarks to the Author)

I have reviewed the revision and am satisfied with the efforts that the authors made to improve the manuscript. I believe they have well addressed the remaining of my concerns and recommend the acceptance of this manuscript for publication.

Review of: “A transient non-soluble noble metal transport in hydrothermal ore systems”

By: Néstor Cano, José M. González-Jiménez, Antoni Camprubí, Eduardo González-Partida

Submitted to: *Nature Communications*

Manuscript Number: NCOMMS-24-33396

Reviewer: Andrew Tomkins, Monash University

Summary

This paper presents an interesting discussion of how epithermal gold deposits, exemplified by very gold-rich samples from the El Hilo deposit, might form by transport and coalescence of nano-sized polymetallic melt droplets. I generally find the paper very interesting and the combination of ideas is plausible. I have published a bit on Au-rich polymetallic melts myself, so my tendency would be to support publication of work on this topic. However, the problem is that there is not any strong evidence presented to support their model of long-distance droplet transport. The work showing that there were droplets of Au-rich polymetallic melt present in the studied samples is well done and the figures are nice. However, that sort of work has been published before and by itself is not new enough to be publishable in *Nature Comms*. The new idea, which could bring the paper to point of being suitable for *Nature Comms*, focus on transport of nanoparticulate polymetallic melt droplets. I find that sort of new idea to be interesting and creative and my tendency is to encourage editors to publish this sort of thing. One problem is that the Stokes Law modelling is too simplistic to be a passably reliable way of representing the system at hand. The discussion that follows is speculative and not supported by any evidence that I consider to be strong enough to support what they are saying. The ideas are plausible, but there really isn't enough basis for putting them forward (see my more detailed comments below). I think that I have to recommend rejection. If the authors could find stronger evidence and undertake more suitable modelling it would be appropriate to resubmit. I have made some suggestions for editing the text on the PDF file and my specific comments are listed below.

Specific Comments

Table S1: There are a lot more than those listed in this table. If you're aiming for a comprehensive list, you have a bit of work to do (e.g., do a quick search on the Au-Bi association). If you're aiming for a list of some examples, this is probably OK, but you should indicate that the list is just an exemplar of a bigger set of known deposits.

Line 46: Careful here. Many people have recently postulated that nano-particulate gold is important in ore formation, so your wording here is inaccurate. It would be better to say that few people have investigated the role of nano-sized melt droplets in ore formation, which is what you're actually doing. Also, be aware that work similar to yours has been published already, so you should be citing that and indicating how

your work is different, which it is (you should do a comprehensive search for more, such as amongst the 42 papers that cite the one I'm listing here):

Hastie, E.C.G., Schindler, M., Kontak, D.J. et al. (2021) Transport and coarsening of gold nanoparticles in an orogenic deposit by dissolution–reprecipitation and Ostwald ripening. *Commun Earth Environ* 2, 57. <https://doi.org/10.1038/s43247-021-00126-6>

Line 59: If you're looking at inclusions trapped in mineralised quartz veins in an ore deposit, you cannot glean insights into the early stages of metal transport. The inclusions are trapped as quartz crystallises, so the thermo-chemical conditions at the time of fluid infiltration are not even preserved well (the conditions have to change in order for quartz to form).

Line 61: "...mechanic[al] transport of metallic nanomelt precursors..." What is this supposed to mean? The precursor to a nanomelt droplet would be an aqueous complex like AuHS⁻ for example, and talking about mechanical transport of that makes no sense. Perhaps just delete the word "precursors"? Or use language that explains what you mean.

Line 119: Yes, but be a little careful here; it applies for nanoparticles < ~5 nm.
See:

Dick et al., 2002, Size-dependent melting of silica-encapsulated gold nanoparticles. *J. Am. Chem. Soc.* 124, 10, 2312–2317.

Lines 120-123: I have no problem with the idea that nanos can sometimes serve as nuclei for growth of bigger things, whether they be minerals or melt droplets. However, I don't think their existence in a mineralised quartz vein, amongst some larger polymetallic melt droplets, is evidence that they formed early. It's just evidence that they existed at the same place and time. It is plausible that they could have formed a long way away and been carried to the site of deposition, but it's probably more likely that they formed as the quartz hosting them crystallised, given that this would have been in response to changing T, P and/or X conditions (X being chemistry). You could delete the over interpretation from here and leave the overall point til later after you've discussed physical carriage.

Fig 3a, b: In *a*, move the numbers down onto the x-axis and perhaps find a better way to indicate the size bins (why not use whole numbers to save space?). In *b*, why is the critical size in the middle of the fluid ascent rate range? Would be more conservative to be at the bottom – that's also similar to the size of many of your particles, which might explain why they're there.

Line 185: "...beyond which they would have sank and been lost." Hydrothermal plumbing systems are not continuous vertical pipes through which things can be carried or sink. They are tortured pathways, commonly with a significant horizontal component, that are activated by fluid-mediated deformation. Yes, small particles can be carried when the flow rate is high enough, but no, particles will not sink; they just will be too heavy to be carried along. See next comment.

General point about the usage of Stokes Law: Using the simple version of Stokes Law for this exercise is a major simplification of all hydrothermal systems, which inherently have complex structurally controlled fluid flow pathways. Commonly the structures that allow fluid transit are activated by addition of fluid from below in a dynamic environment (increased pore fluid pressure causes the rocks to break, allowing fluid transmission). The simple version of Stokes Law applies to a continuous body of fluid, such as a vertical pipe with frictionless walls, that dense particles sink through; there is no tortuosity in the migration path. Those working on particle sinking through melt networks in a solid framework of crystals have derived modifications of Stokes Law to deal with the tortuosity problem. But this still wouldn't apply directly to your case. Perhaps somebody has already attempted to numerically model carriage of dense particles in a hydrothermal fluid system?

Currently, Figure 3b and the discussion surrounding it are largely irrelevant because of this over simplification. The discussion on whether a given size particle can be transported is not based on appropriate modelling. Are the fluid ascent rates given an average of the ascent rate through a broad section of crust, or are they local ascent rates applicable to a particular structure?

One possible way to deal with this problem is to acknowledge the difficulty of modelling a complex system, show that the simple Stokes Law modelling finds that sub-micron particles settle at rates 3 orders of magnitude slower than estimated fluid ascent rates (Fig 3b), and therefore suggest that nanodroplets are likely to be easily transported long distances in hydrothermal systems (not the cautious tone of the text). The other way to deal with the problem is to numerically model the complex system, which is perhaps what's needed for a *Nature Comms* paper.

Lines 185-189: "Recent studies^{8,9,15,27} have shown that shallow boiling, cooling, and mixing with external fluids (such as meteoric water or seawater) promote the coalescence of nano-sized materials in hydrothermal fluids. These processes are characteristic of the shallow epithermal environment^{28,29}, suggesting that some polymetallic melts from El Hilo **should** have originated outside this environment." Sure, coalescence is likely promoted in that environment, but this doesn't require that the particles formed outside that environment. Given the dramatic changes in P, T and X, it's more likely that they formed when the fluids experienced boiling, cooling, mixing. I agree that it's plausible that some of them formed prior to reaching the site of deposition.

Response to the comments on the manuscript:

A transient non-soluble noble metal transport in hydrothermal ore systems

Néstor Cano, José M. González-Jiménez, Antoni Camprubí, Eric Morales-Casique, and Eduardo González-Partida

For practicality, the lines referenced throughout this document correspond to those in the Manuscript_no tracked changes file, which is included as a related manuscript file.

Response to the comments by Reviewer 1

(Reviewer's comments are italicized and our responses are in blue)

We thank Reviewer 1 for taking the time to revise our manuscript and for their overall positive appraisal. Their comments have considerably improved the article. We have responded to the suggested changes point-by-point below.

The manuscript presents an intriguing study on the transport of noble metals by polymetallic melts in hydrothermal ore systems, proposing this as a complementary mechanism to metal-bearing ligands in solution. The authors provide evidence for suspension-like transport of nano-to-micron sized polymetallic melts within epithermal fluids, leading to the formation of polymetallic inclusions in quartz. While the manuscript introduces exciting ideas, there are some areas that require further clarification and discussion.

Major Issues

Data Interpretation:

The authors interpret the polymetallic inclusions in quartz as indicative of deposition from a melt (line 110). Since this textural interpretation forms the core of their model, it is essential that it is robust and well-supported. It is possible that these textures could also result from the sudden deposition of dissolved metals and sulphur from hydrothermal fluids during a boiling event. To strengthen their interpretation, the authors should provide a more thorough discussion, demonstrating that no other deposition processes could account for the observed textures.

We agree with this comment and have included a more complete discussion in lines 110–128. Indeed, boiling (cooling, neutralization, etc.) effectively triggers deposition of metals in solution, probably forming similar textures as those shown in Figs. 1 and 2. However, El Hilo is a ~5 cm-wide bonanza ore with very high Au and Ag contents (2 kg/t Au and 31 kg/t Ag), which lacks any laminated arrangement suggestive of repeated fluid injections. In addition, the extent of hydrothermal veining and alteration haloes at El Hilo are comparable to those of neighboring less mineralized veins, suggesting similar amounts of circulating fluids. We thus interpret that El Hilo records a single hydrothermal pulse that was over-enriched in Au and Ag.

Such a precious metal focus is difficult to reconcile with *in-situ* precipitation from dissolved species alone, considering the solubility limits for these metals in hydrothermal solutions (hundreds of ppb; e.g., Pokrovski et al., 2014; Pearce et al., 2015; Wagner et al., 2016; Liu et al., 2019). To explain this inconsistency, ours and previous studies have proposed complementary mechanisms, including the involvement of Au-Ag colloidal suspensions (e.g., Saunders and Burke, 2017; McLeish et al., 2021, 2024; Petrella et al., 2022) or (S-bearing) metallic melts rich in low-melting-point chalcophile elements (LMCE, As, Sb, Se; e.g., Ciobanu et al., 2006; Jian et al., 2022, 2024; Vikent'eva and Bortnikov, 2023). The occurrence of LMCE + precious metal-bearing mineral assemblages in polymineral inclusions suggests that metallic melts are a more plausible mechanism than Au-Ag colloidal suspensions to explain the El Hilo mineralization.

Upon reviewing the provided figures (Figure 1.f, g, h, i, j, and Figure 2.a and b), the polymetallic inclusions do not appear particularly blebby. More evidence and detailed explanation are needed to substantiate the claim that these textures are indeed a result of deposition from a melt.

We agree that some inclusions are irregular instead of blebby. However, this is particularly evident in the largest inclusions from Figs. 1f–j and Supplementary Fig. S2. In contrast, most smaller inclusions (e.g., Figs. 1e, 2b–m, S2) appear blebby at the micron and nanoscale, some even being perfectly rounded (e.g., Fig. 2e and g). We believe this results from the coalescence and growth of metallic melts upon migration (or deposition), which agrees with the notable compositional and textural similarities between micro- and nano-sized inclusions (lines 220–223). It is worth noting that polymineral inclusions (PMI) with comparable shapes and textures—although larger than those at El Hilo—have been recently proved by Jian et al. (2024) to be former metallic melts. These authors performed heating experiments and induced the partial melting of PMIs at hydrothermal temperatures (<400 °C).

Composition Discrepancy:

The abstract states that nano-to-micron sized polymetallic melts within epithermal fluids form bleb-like polymetallic inclusions of Au-Ag(-As-Se-Sb) (line 24). However, the data presented in the manuscript indicate that the polymetallic inclusions are composed of "Ag(-Cu)-Sb-As-Se sulfosalts, electrum, and Ag-Cu-Fe-Pb sulfides." Furthermore, line 108 describes a "Ag-Au-Cu-Pb(-Fe-Zn)-As-Sb-S-Se multi-component system." The presence of sulphur is significant as it influences the behaviour of the polymetallic melt. This inconsistency between the abstract and the actual data presented needs to be addressed to maintain the reliability of the conclusions drawn.

Agreed. We have corrected these issues in lines 23, 80, 115, and 130–131.

Transport of PM Droplets:

The manuscript suggests that polymetallic (sulfosalt and sulfides) material is mechanically transported by rising hydrothermal fluids as melt droplets (lines 182–184, 191–193) from a "distally located" magmatic intrusion to the surface. While the authors demonstrate that velocity calculations make such transport possible, it is not discussed how such material would remain in a chemically stable suspension within the hydrothermal fluid. Existing experimental studies show that sulfide melts are not easily transported (Rose and Brenan, 2001).

Agreed. Previous workers have found that sulfide and polymetallic melts are able to migrate short distances (centimeters to meters) during prograde metamorphism (Frost et al., 2002; Tomkins et al., 2007) or magmatic crystallization (Rose and Brenan, 2001). However, these workers have focused on metallic melt mobility occurring along grain boundaries, in the absence of an external transporting agent (e.g., fluid or magma). In our case study, we assess the mobility of nanomelt droplets by upwelling hydrothermal fluids, which would permit greater migration distances than those achieved by metallic melts alone. Thus, it would be inadequate to compare both scenarios. Indeed, some authors have found that sulfide melts can be physically transported across crustal-scale distances by silicate magmas in some tectonic settings (e.g., Schettino et al., 2023).

The manuscript does not sufficiently address the chemical processes by which the melt droplet could be kept in a stable suspension in the aqueous epithermal fluid ("LMCE-bearing metallic (nano)melts could be kept suspended," line 233). The potential for the melt to react with the aqueous fluid, rather than remaining inert and stable, needs more discussion.

We agree with this comment and have included lines 210–223 to support our ideas. We think that metallic melts did not remain inert but, instead, could have reacted with the fluid by (1) scavenging metals from dissolved species, and/or (2) being partly dissolved by the fluid. On the one hand, precious metal scavenging by metallic melts could have occurred during migration (or upon deposition) and agrees with the very high Au and Ag contents of PMIs (2 wt% Au and 61 wt% Ag)

and, naturally, of the El Hilo bonanza (2 wt% Au and 31 wt% Ag). This is explained by the preferential partitioning of metals from aqueous solutions to metal-rich sulfide melts (Tooth et al., 2008, 2011).

On the other hand, available solubility models for hydrothermal fluids at 10 wt% NaCl equiv., pH ~5, and 350 °C—which are conditions similar to those of El Hilo fluids—yield solubility limits at ~0.1 ppm Au and 50 ppm Ag₂S (Kouzmanov and Pokrovski, 2012). These data align with calculations by other authors, which suggest Au solubilities on the order of ppbs (e.g., Seward et al., 2014; Pearce et al., 2015; Wagner et al., 2016; Liu et al., 2019). Considering this, dissolution of Au and Ag amounts as high as those found at El Hilo (in the wt% scale) would have required large volumes of hydrothermal fluids, which are not reflected in the extent of quartz veining and/or alteration haloes at El Hilo when compared to neighboring low-grade or unmineralized stages of mineralization (lines 120–128). Thus, it is likely that ore-bearing fluids of El Hilo had met their solubility limits and were unable to dissolve any more precious metals from suspended material without attaining their saturation and precipitation. Consequently, we think that an at-large dissolution of metallic melts by the fluids has no logical support (lines 212–216).

Additionally, how the nano-to-micro size droplets would percolate through kilometres of rock without reacting with the wall rock or being mechanically stopped by limited porosity requires further elaboration. To demonstrate the transport of the melt, the authors should explain how it can be in both mechanical and chemical equilibrium within the fluid.

We agree with this comment. Although the location/depth of the magmatic source at Natividad is unknown, intermediate-sulfidation epithermal deposits typically occur <4 km away from their causative intrusion (Camprubí and Albinson, 2007; Sillitoe, 2010; Wilkinson et al., 2013). Veins at Natividad are structurally controlled by NE-trending fault systems, which produce fracture networks that constitute efficient fluid paleo-conduits (Cano et al., 2024). It is likely that these paleo-conduits facilitated the upwelling of hydrothermal fluids by reducing the chance of being stopped by limited porosity lithologies. Although we cannot assure that the fluid flow was continuous (and rapid) from the source to the deposition site, our model (Fig. 3b–c) shows that metal-rich sulfide nanomelts could have been transported even at very low flow rates (<0.01 m/s). If the fluids were indeed stopped at some point during ascent, so that nanoPMIs settled, Fig. 3b shows that the addition of more fluid from below at a given ascent rate (<10⁻⁵ m/s) would have readily fluidized the particles again.

As for possible reactions with the wall rocks, we have found no explanations or cues from the existing literature as to how these would have been possible. Therefore, we chose not to speculate on the matter and, instead, referred to the plausibility of reactions with the hydrothermal fluid itself, either by partial solubilization by the fluid or by elemental scavenging by the droplets (lines 210–223). Moreover, we agree that melt droplets near the walls of the fractures could have been physically trapped (e.g., sorption or obstruction) by the wall rocks. However, those along the center of the flow were more likely to have been fluidized and transported, aided by the connectivity of the fracture network.

Specific Comments

Line 24: The statement about the composition of polymetallic inclusions does not match the data presented. The manuscript needs to clarify that the inclusions are primarily sulfide melts, which alters the interpretation of the findings.

Agreed. We have replaced “polymetallic” with “metal-rich sulfide melts” throughout the text.

Line 34: The references cited (Econ. Geol. 99, 1063–1084 (2004); Toote 2008 and 2011; Frost 2002) are used to support the hypothesis of polymetallic melts in hydrothermal fluids. However, these studies primarily discuss local migration within host rocks or low-melting point chalcophile elements, not

the long-distance transport in aqueous fluids proposed by the authors. The manuscript should critically assess these references and clarify their relevance to the current study.

Although we agree with this comment, we used the references (Frost et al., 2002; Tomkins et al., 2004; Tooth et al., 2008, 2011) in the sense that their research proved that metal-rich sulfide and polymetallic melts could exist at hydrothermal conditions (temperature, fS_2 , fO_2) and can even exsolve directly from hydrothermal fluids. We do not use them to discuss melt transport within the fluids. We have rephrased lines 33–36 for clarity. Also, we refer to thermodynamic calculations of Frost et al. (2002) and Tomkins et al. (2004) to show that metal-rich sulfide melts were stable at fluid temperature ranges established for the El Hilo bonanza ores (i.e., fluid temperatures up to 400 °C). Long-distance transport as such is addressed by means of our calculations and the works of Oliver et al. (2006) and Okamoto and Tsuchiya (2009).

Line 78: There is a critical need for a more detailed discussion on the phases referred to as PMI by the authors: Ag(-Cu)-Sb-As-Se sulfosalts, electrum, and Ag-Cu-Fe-Pb sulfides. The manuscript must differentiate between polymetallic melts and sulfide melts and address their transport mechanisms individually.

Agreed. We have replaced “polymetallic inclusion” with “polymineral inclusions” to be more precise. We did not find any S-free inclusion (except for some electrum inclusions), as PMIs average 15.5 ± 4.9 wt% S (1σ , Table S2).

Line 105: More detailed descriptions of the textures are needed to convince the reader that the textures are indeed the remnant of melts.

Agreed. We have included some additional information on textures in lines 110–117. Nearly identical textures and compositions of PMIs have been interpreted by several authors (cf. Ciobanu et al., 2006; Sousa-Guimarães et al., 2019; Jian et al., 2022, 2024; Vikent’eva and Bortnikov, 2023) as evidence of deposition from a melt in a range of ore deposits.

Line 122: Can you be more specific here regarding what early segregation refers to? Do you mean early segregation from a parental magma?

We meant segregation from the source(s), which could have been the parental magma and/or hydrothermal fluids themselves. We deleted “segregation” and rephrased this line to be more precise.

Line 131: This statement is slightly confusing. It does not appear that your data shows sulfides PMI within a fluid inclusion. Rather, they seem to be occurring within the same quartz crystal.

Agreed. We did not observe any polymineral inclusions (PMI) inside individual fluid inclusions. Instead, PMI inclusions occur with fluid inclusion assemblages (FIA) in quartz. That is, PMIs are considered as solid inclusions (as is the specific term in the common lore of fluid-inclusionists) with which fluid inclusions were co-trapped. Such is the most logical interpretation as per the evidence provided through the specific petrographic studies that are addressed to fluid inclusions. We have rephrased lines 147–150 for clarity.

Line 169-170: This sentence needs clarification. It is unclear what is interpreted to happen to metals in suspension and how it relates to polymetallic melt.

Agreed, we have reformulated lines 224–227. We propose that fluid turbulence caused by boiling increased the Brownian motion of suspended melts, thus enhancing their reaction (i.e., scavenging of metals) with the ore-bearing fluid and their coalescence.

Line 191-193: This part needs more explanation. Are the polymetallic melts at that stage sulfide melts? (It should be, considering your data.)

We agree with this comment. Assuming that sulfide melts were formed in the causative intrusion (this is one possibility; lines 242–256 and 271–286), we do think that they should have been metal-rich

sulfide melts at the stage of magmatic fluid release. This hypothesis agrees with $\delta^{34}\text{S}_{\text{VCDT}}$ between -3.2% and -0.3% in sulfides and sulfosalts from El Hilo (Cano et al., 2024), matching the range of mantle S (Chaussidon and Lorand, 1990; line 242-244). It also agrees with previous reports of metal-rich sulfide melts in porphyry(-epithermal) systems (e.g., Elatsite, Bulgaria; González-Jiménez et al., 2021), supporting the idea that metal-rich sulfide+LMCE melts can be produced during deep magmatic fluid release (lines 271–286).

More discussion is needed to explain how the sulfide melts/droplets can be segregated from the magma into the aqueous fluid.

We agree with this comment and have added lines 273–283 accordingly. Metals behave incompatibly during magmatic differentiation and are thus concentrated alongside volatiles until the late stages of crystallization. Once volatile saturation is achieved, the magmatic fluid phase (+ metals) exsolves and migrates to produce (or not) mineralization (Becker et al., 2019; and references therein). Considering the metal endowment of the exsolved fluids and their temperature ranges (up to 650 °C; Kouzmanov and Pokrovski, 2012)—which are above the eutectic point of several LMCE-bearing chemical systems—, metal-rich melts could potentially segregate directly from these fluids (e.g., Tooth et al., 2011).

In our view, sulfide melt droplets did not necessarily coexist with the magma before being segregated into the fluid phase. For instance, the results of Schettino et al. (2023) suggest that sulfide melts can occur in magmas and be transported (mechanically) by them from the mantle to shallow magma reservoirs. Once in shallow reservoirs, Jeanvoine et al. (2024) found that sulfide melt droplets can well up across magmas and eventually transfer metals to the hydrothermal fluid via flotation (lines 276–279). These authors showed that Cu-rich (+Pd+Au) sulfide droplets $<65\ \mu\text{m}$ in diameter could be transported upward by water bubbles, and subsequently supply metals to form Cu-Au-rich seafloor sulfide deposits in subduction settings. Although this combination of processes could have occurred near the magmatic source of El Hilo, we think that it was not a necessary prerequisite for the formation of bonanza-type ores because metal-rich sulfide melts could have also been produced directly from the fluids (Tooth et al., 2011) along the way to the deposition site (this is the other possibility; lines 296–298).

Line 232-233: There is a discrepancy between the literature cited and the interpretation made by the authors.

Agreed. We have rephrased lines 273–276.

The metal-rich brines referred to in line 232 are brines in which metals are dissolved as chloro-complexes and not suspended as nanomelts. There is no data to support the claim that nanomelts are present in the brines.

It is arguable that the entire metal load may come dissolved as chloride complexes. Firstly, it is broadly known that gold chloride complexes would not be stable in this environment, and it would be likely transported as thiosulfide complexes, rather than chlorides. Such complexes are assumed to be subordinate in intermediate sulfidation brines, provided that most intermediate sulfidation epithermal deposits worldwide tend to be depleted in gold, compared to their low and high sulfidation counterparts. Therefore, the exceptionally high gold grades in this mineralized section may additionally underpin the nanomelt hypothesis as an essential contributor of this metal. We do not discuss this issue in the manuscript because it stems as speculative and, however reasonable, there is no way to prove it. As for the capability of relatively high salinity brines (as determined from fluid inclusion data) for transporting silver and base metals, it is beyond doubt, as is common lore from epithermal deposits already.

Secondly, the geological, geochemical, and textural evidence obtained at El Hilo does not support that the metal load was entirely provided by soluble metal-bearing complexes. This bonanza zone is a few cm wide and contains weight percents of Au and Ag, most of which occur as micro- to nano-sized polymineral inclusions within quartz. As discussed earlier, the extent of hydrothermal veining at El Hilo does not justify the amount of fluid required to explain these grades by precipitation from dissolved species alone (e.g., AgCl_2^- , AuCl_2^- , and/or AuHS_2^-)—considering precious metal solubility limitations (Pokrovski et al., 2014; Pearce et al., 2015; Wagner et al., 2016; Liu et al., 2019). These observations support a transient suspension-like (i.e., colloidal) transport of nanomaterials within the fluids, which agrees with our numerical models (Fig. 3b–c). Of course, direct evidence of nanomelts in inclusion fluids, which would be an iron-clad proof for their occurrence, was out of our reach. However, in our view, and in accordance with preexisting literature, we have provided evidence that is credible enough as to account for the occurrence of such nanomelts.

A discussion is needed to address how such nanomelts would be stable in this environment and how they might have been extracted from the magmatic precursor.

Agreed. Constraints on nanomelt stability and extraction from the magmatic precursor are provided in lines 271–283.

Figure 1: The manuscript lacks a discussion on quartz dissolution textures presented in Figure 1, which could provide insights into the deposition mechanisms and the environment of formation.

Agreed. We have double-checked our samples and expanded the discussion on quartz textures in lines 81–85. In addition, we have included some photomicrographs of quartz dissolution-replacement textures in the new Supplemental Fig. S1.

Figure 4: The composition of the rising fluid and its ability to stabilise polymetallic inclusions during transport must be addressed.

We agree with this comment and have included lines 210–223 accordingly. In addition, we have modified Fig. 4a to provide a more complete description of our model. It would be impossible to reliably constrain the composition of rising ore-bearing fluids prior to their trapping in fluid inclusions. However, our fluid inclusion data show that mineralizing fluids have temperatures up to ~400 °C and salinities up to ~19 wt% NaCl equiv. (Cano et al., 2023). In the absence of evaporites around Natividad (and regionally), these salinity ranges suggest a remarkable component of magmatic brines in the fluids. Consequently, it is likely that the upwelling ore-bearing fluid had a similar (or somewhat less diluted) composition. Regardless of the composition of the fluid input, it could have not dissolved all Au and Ag contained in suspended materials to form the 5 cm-wide bonanza zone, given the metal solubility limitations that have been explained earlier. We argue that this could have favored the stabilization of metal nanomaterials instead of their dissolution.

This includes discussing the role of fluid-rock interactions and the impact of wall rock porosity.

We agree with this comment and have included lines 287–305 to discuss these factors. Hydrothermal veins at the Natividad deposit are hosted by porphyritic dacites and/or carbonaceous metasedimentites (Fig. 4a). However, the highest Au grades occur in those veins (including El Hilo) hosted by the latter. Thus, it is likely that redox reactions between oxidized mineralizing fluids and reduced carbonaceous materials could have been an ingredient in focusing precious metals (lines 287–298). In agreement, we previously found methane in some fluid inclusions from El Hilo (Cano et al., 2024)—probably formed via thermogenesis of organic matter present in wall rocks—, which provides evidence for fluid-rock interaction to some extent. Although we cannot assert to what extent these reactions contributed to ore formation, we can hypothesize their role as observed by authors in other hydrothermal ore deposits (e.g., Petrella et al., 2021).

As for the impact of wall-rock porosity, veins at Natividad are structurally controlled by faults that constitute fluid paleo-conduits (Cano et al., 2024). These paleo-conduits could have enhanced the upwelling of hydrothermal fluids by reducing the chance of being hindered or stopped by limited porosity lithologies. Nevertheless, if the fluids were indeed stopped at some point during ascent so that the nanoPMIs settled, Fig. 3b shows that the addition of more fluid from below at a given ascent rate ($<10^{-5}$ m/s) would have readily fluidized the particles again. As additional information (not included in this manuscript or in earlier publications), we may also state that we found neither lithologies that came to disperse fluid flow laterally nor any broadening of hydrothermal alteration halos that would be suggestive of such a case. In fact, alteration halos around the mineralized structures in metasedimentites are so narrow that they are barely visible at a macroscopic (and even microscopic; Cano et al., 2024). Therefore, although we cannot know the extent of water-rock interaction below the known mineralized area, it seems unlikely that such a mechanism can be regarded as a ruling factor for chemical reactions other than very locally-driven redox reactions.

CITED REFERENCES

- Becker, S.P., Bodnar, R.J., and Reynolds, T.J., 2019, Temporal and spatial variations in characteristics of fluid inclusions in epizonal magmatic-hydrothermal systems: Applications in exploration for porphyry copper deposits: *Journal of Geochemical Exploration*, v. 204, p. 240–255, doi:10.1016/j.gexplo.2019.06.002.
- Camprubí, A., and Albinson, T., 2007, Epithermal deposits in México - Update of current knowledge and an empirical reclassification: *Special Paper of the Geological Society of America*, v. 422, p. 377–415, doi:10.1130/2007.2422(14).
- Cano, N., Camprubí, A., Proenza, J.A., and González-Partida, E., 2024, Genesis and evolution of the Natividad Au-Ag(-Ge) epithermal deposit in southern Mexico: Unravelling the final chapters of the Sierra Madre del Sur metallogeny: *Ore Geology Reviews*, doi:https://doi.org/10.1016/j.oregeorev.2024.105979.
- Cano, N., González-Jiménez, J.M., Camprubí, A., Domínguez-Carretero, D., González-Partida, E., and Proenza, J.A., 2023, Nanomaterial accumulation in boiling brines enhances epithermal bonanzas: *Scientific Reports*, v. 13, p. 1–7, doi:10.1038/s41598-023-41756-4.
- Chaussidon, M., and Lorand, J.P., 1990, Sulphur isotope composition of orogenic spinel lherzolite massifs from Ariège (North-Eastern Pyrenees, France): An ion microprobe study: *Geochimica et Cosmochimica Acta*, v. 54, p. 2835–2846, doi:10.1016/0016-7037(90)90018-G.
- Ciobanu, C.L., Cook, N.J., Damian, F., and Damian, G., 2006, Gold scavenged by bismuth melts: An example from Alpine shear-remobilizates in the Highiş Massif, Romania: *Mineralogy and Petrology*, v. 87, p. 351–384, doi:10.1007/s00710-006-0125-9.
- Frost, B.R., Mavrogenes, J.A., and Tomkins, A.G., 2002, Partial melting of sulfide ore deposits during medium- and high-grade metamorphism: *Canadian Mineralogist*, v. 40, p. 1–18, doi:10.2113/gscanmin.40.1.1.
- González-Jiménez, J.M., Piña, R., Kerestedjian, T.N., Gervilla, F., Borrajo, I., Pablo, J.F. de, Proenza, J.A., Tornos, F., Roqué, J., and Nieto, F., 2021, Mechanisms for Pd–Au enrichment in porphyry-epithermal ores of the Elatsite deposit, Bulgaria: *Journal of Geochemical Exploration*, v. 220, p. 106664, doi:10.1016/j.gexplo.2020.106664.
- Jeanvoine, A., Park, J.W., Pelleter, E., Bézos, A., Chazot, G., Hwang, J., and Fouquet, Y., 2024, The flotation of magmatic sulfides transfers Cu-Au from magmas to seafloor massive sulfide deposits: *Communications Earth and Environment*, v. 5, p. 1–16, doi:10.1038/s43247-024-01571-9.
- Jian, W., Mao, J., Cook, N.J., Chen, L., Xie, G., Xu, J., Song, S., Hao, J., Li, R., and Liu, J., 2022, Intracrystalline migration of polymetallic Au-rich melts in multistage hydrothermal systems:

- example from the Xiaqingling lode gold district, central China: *Mineralium Deposita*, v. 57, p. 147–154, doi:10.1007/s00126-021-01090-z.
- Jian, W., Mao, J., Lehmann, B., Cook, N.J., Li, J., Song, S., and Zhu, L., 2024, Hyper-enrichment of gold via quartz fracturing and growth of polymetallic melt droplets: *Geology*, v. 52, p. 411–416, doi:10.1130/G51875.1/6275576/g51875.pdf.
- Kouzmanov, K., and Pokrovski, G.S., 2012, Hydrothermal Controls on Metal Distribution in Porphyry Cu (-Mo-Au) Systems: Society of Economic Geologists, Special publication, v. 16, p. 573–618, doi:10.5382/sp.16.22.
- Liu, W., Chen, M., Yang, Y., Mei, Y., Etschmann, B., Brugger, J., and Johannessen, B., 2019, Colloidal gold in sulphur and citrate-bearing hydrothermal fluids: An experimental study: *Ore Geology Reviews*, v. 114, p. 103142, doi:10.1016/j.oregeorev.2019.103142.
- McLeish, D.F., Williams-Jones, A.E., Clark, J.R., and Stern, R., 2024, Extreme shifts in pyrite sulfur isotope compositions reveal the path to bonanza gold: *Proceedings of the National Academy of Sciences*, v. 121, doi:https://doi.org/10.1073/pnas.2402116121.
- McLeish, D.F., Williams-Jones, A.E., Vasyukova, O. V., Clark, J.R., and Board, W.S., 2021, Colloidal transport and flocculation are the cause of the hyperenrichment of gold in nature: *Proceedings of the National Academy of Sciences of the United States of America*, v. 118, p. 1–6, doi:10.1073/pnas.2100689118.
- Pearce, M.A., White, A.J.R., Fisher, L.A., Hough, R.M., and Cleverley, J.S., 2015, Gold deposition caused by carbonation of biotite during late-stage fluid flow: *Lithos*, v. 239, p. 114–127, doi:10.1016/j.lithos.2015.10.010.
- Petrella, L., Thébaud, N., Evans, K., LaFlamme, C., and Occhipinti, S., 2021, The role of competitive fluid-rock interaction processes in the formation of high-grade gold deposits: *Geochimica et Cosmochimica Acta*, v. 313, p. 38–54, doi:10.1016/j.gca.2021.08.024.
- Petrella, L., Thébaud, N., Fougereuse, D., Tattitch, B., Martin, L., Turner, S., Suvorova, A., and Gain, S., 2022, Nanoparticle suspensions from carbon-rich fluid make high-grade gold deposits: *Nature Communications*, v. 13, p. 1–9, doi:10.1038/s41467-022-31447-5.
- Pokrovski, G., Akinfiyev, N., Borisova, A., Zotov, A., and Kouzmanov, K., 2014, Gold speciation and transport in geological fluids: insights from experiments and physical-chemical modelling, in Garofalo, P.S. and Ridley, J.R. eds., *Gold-transporting hydrothermal fluids in the Earth's crust*, London, Geological Society of London, p. 9–70, doi:https://doi.org/10.1144/SP402.4.
- Rose, L.A., and Brenan, J.M., 2001, Wetting properties of Fe-Ni-Co-Cu-O-S melts against olivine: Implications for sulfide melt mobility: *Economic Geology*, v. 96, p. 145–157, doi:10.2113/gsecongeo.96.1.145.
- Saunders, J.A., and Burke, M., 2017, Formation and aggregation of gold (Electrum) nanoparticles in epithermal ores: *Minerals*, v. 7, p. 1–11, doi:10.3390/min7090163.
- Schettino, E., González-Jiménez, J.M., Marchesi, C., Palozza, F., Blanco-Quintero, I.F., Gervilla, F., Braga, R., Garrido, C.J., and Fiorentini, M., 2023, Mantle-to-crust metal transfer by nanomelts: *Communications Earth and Environment*, v. 4, p. 1–9, doi:10.1038/s43247-023-00918-y.
- Seward, T.M., Williams-Jones, A.E., and Migdisov, A.A., 2014, The chemistry of metal transport and deposition by ore-forming hydrothermal fluids, in Holland, H. and Turekian, K.K. eds., *Treatise on Geochemistry: Second Edition*, Oxford, Elsevier Ltd., p. 29–57, doi:10.1016/B978-0-08-095975-7.01102-5.
- Sillitoe, R.H., 2010, Porphyry copper systems: *Economic Geology*, v. 105, p. 3–41, doi:10.2113/gsecongeo.105.1.3.
- Sousa-Guimarães, F., Cabral, A.R., Lehmann, B., Rios, F.J., Ávila, M.A.B., de Castro, M.P., and Queiroga, G.N., 2019, Bismuth-melt trails trapped in cassiterite–quartz veins: *Terra Nova*, v. 31, p. 358–365, doi:10.1111/ter.12391.
- Tomkins, A.G., Pattison, D.R.M., and Frost, B.R., 2007, On the initiation of metamorphic sulfide anatexis: *Journal of Petrology*, v. 48, p. 511–535, doi:10.1093/petrology/egl070.

- Tomkins, A.G., Pattison, D.R.M., and Zaleski, E., 2004, The Hemlo Gold Deposit, Ontario: An example of melting and mobilization of a precious metal-sulfosalt assemblage during amphibolite facies metamorphism and deformation: *Economic Geology*, v. 99, p. 1063–1084, doi:10.2113/gsecongeo.99.6.1063.
- Tooth, B., Brugger, J., Ciobanu, C., and Liu, W., 2008, Modeling of gold scavenging by bismuth melts coexisting with hydrothermal fluids: *Geology*, v. 36, p. 815–818, doi:10.1130/G25093A.1.
- Tooth, B., Ciobanu, C.L., Green, L., and Neill, B.O., 2011, Bi-melt formation and gold scavenging from hydrothermal fluids : An experimental study: *Geochimica et Cosmochimica Acta*, v. 75, p. 5423–5443, doi:10.1016/j.gca.2011.07.020.
- Vikent'eva, O. V., and Bortnikov, N.S., 2023, Evidence of Mineral Melting in Ores of the Svetlinsk Gold Deposit, South Urals, Russia: *Doklady Earth Sciences*, v. 513, p. 1321–1325, doi:10.1134/S1028334X2360233X.
- Wagner, T., Fusswinkel, T., Wälle, M., and Heinrich, C., 2016, Microanalysis of fluid inclusions in crustal hydrothermal systems using laser ablation methods: *Elements*, v. 12, p. 323–328.
- Wilkinson, J.J., Simmons, S.F., and Stoffell, B., 2013, How metalliferous brines line Mexican epithermal veins with silver: *Scientific Reports*, v. 3, p. 1–7, doi:10.1038/srep02057.

**Response to the comments by Reviewer 2 (Dr. Andrew Tomkins)
(Reviewer's comments are italicized and our responses are in blue)**

This paper presents an interesting discussion of how epithermal gold deposits, exemplified by very gold-rich samples from the El Hilo deposit, might form by transport and coalescence of nano-sized polymetallic melt droplets. I generally find the paper very interesting and the combination of ideas is plausible. I have published a bit on Au-rich polymetallic melts myself, so my tendency would be to support publication of work on this topic. However, the problem is that there is not any strong evidence presented to support their model of long-distance droplet transport. The work showing that there were droplets of Au-rich polymetallic melt present in the studied samples is well done and the figures are nice. However, that sort of work has been published before and by itself is not new enough to be publishable in Nature Comms. The new idea, which could bring the paper to point of being suitable for Nature Comms, focus on transport of nanoparticulate polymetallic melt droplets. I find that sort of new idea to be interesting and creative and my tendency is to encourage editors to publish this sort of thing. One problem is that the Stokes Law modeling is too simplistic to be a passably reliable way of representing the system at hand. The discussion that follows is speculative and not supported by any evidence that I consider to be strong enough to support what they are saying. The ideas are plausible, but there really isn't enough basis for putting them forward (see my more detailed comments below). I think that I have to recommend rejection. If the authors could find stronger evidence and undertake more suitable modeling it would be appropriate to resubmit. I have made some suggestions for editing the text on the PDF file and my specific comments are listed below.

We thank Dr. Tomkins for taking the time to revise our manuscript! We also appreciate his interest in the topic being discussed in our paper and the insightful comments. We have made all the minor corrections from the attached pdf file and responded to his comments point-by-point below.

Specific Comments

Table S1: There are a lot more than those listed in this table. If you're aiming for a comprehensive list, you have a bit of work to do (e.g., do a quick search on the Au-Bi association). If you're aiming for a list of some examples, this is probably OK, but you should indicate that the list is just an exemplar of a bigger set of known deposits.

Agreed. This list summarizes some examples (line 310 and Table S1).

Line 46: Careful here. Many people have recently postulated that nano-particulate gold is important in ore formation, so your wording here is inaccurate. It would be better to say that few people have investigated the role of nano-sized melt droplets in ore formation, which is what you're actually doing. Also, be aware that work similar to yours has been published already, so you should be citing that and indicating how your work is different, which it is (you should do a comprehensive search for more, such as amongst the 42 papers that cite the one I'm listing here HASTIE 2021):

Agreed! We thank Dr. Tomkins for the recommended bibliography. We have rephrased lines 44–46.

Line 59: If you're looking at inclusions trapped in mineralised quartz veins in an ore deposit, you cannot glean insights into the early stages of metal transport. The inclusions are trapped as quartz crystallises, so the thermo-chemical conditions at the time of fluid infiltration are not even preserved well (the conditions have to change in order for quartz to form).

Yes and no. Fluid inclusion studies constitute a reasonable approach to thermochemical conditions of mineral deposition but, for various reasons (which may lead to a long dissertation), they are never 100% beyond reasonable doubt. Serious fluid inclusionists aim to ensure the best possible way to perform representative analyses through scrupulous petrographic and microthermometric

determinations, which we did (see Cano et al., 2023, 2024). Our methodological procedure on the matter was canonical, and aimed to overcome the fact that, as we all know, fluid inclusions sample actual mineralizing fluids in a discontinuous and somewhat serendipitous manner. In this sense, fluid inclusion studies have similar limitations as the studies on the fossil record, but a discontinuous record does not invalidate any study in which all due precautions were taken, and all methodological or instrumental good practices were employed.

We agree that if all metals were being transported in solution, then the observed polymineral inclusions should have precipitated *in-situ* as mineralized quartz was growing in response to physicochemical variations in ore-bearing fluids (as Reviewer mentioned). Direct precipitation of focalized Au-Ag ores grading in the wt% scale found at El Hilo would have required episodic injections of large volumes of hydrothermal fluid. However, this ~5 cm-wide bonanza zone lacks any structure suggestive of episodic fluid percolation (e.g., colloform-banded or crustiform; Fig. 1a), which points to a single metal-rich fluid injection. Also, the extent of hydrothermal veining at El Hilo is comparable to that around neighboring less mineralized veins, suggesting similar volumes of circulating fluids. Alternatively, precipitating such amounts of Au-Ag from a single metal-rich fluid injection would have required unrealistically high Au-Ag solubilities (lines 117–125). These observations suggest that part of Au-Ag-bearing nanomaterials were probably already present within the fluid at the onset of *in-situ* quartz + metal precipitation. In this sense, these “carried-away” nanomaterials would represent an “early” stage of metal transport.

Line 61: “...mechanic[al] transport of metallic nanomelt precursors...” What is this supposed to mean? The precursor to a nanomelt droplet would be an aqueous complex like AuHS- for example, and talking about mechanical transport of that makes no sense. Perhaps just delete the word “precursors”? Or use language that explains what you mean.

Agreed. We have corrected “precursors” to “droplets”.

Line 119: Yes, but be a little careful here; it applies for nanoparticles < ~5 nm. See: Dick et al., 2002, Size-dependent melting of silica-encapsulated gold nanoparticles. J. Am. Chem. Soc. 124, 10, 2312–2317.

We thank Dr. Tomkins for the recommended paper. We have reformulated lines 137–139 to be more precise. According to Guisbiers et al. (2016), the melting point of Au-Ag alloys is size- and shape-dependent and decreases as Ag contents increase—even in nanoparticles up to 10 nm in diameter. Provided that some of the observed nanoparticles reside in this size interval, we think that the size did facilitate the formation of nano-droplets.

Lines 120-123: I have no problem with the idea that nanos can sometimes serve as nuclei for growth of bigger things, whether they be minerals or melt droplets. However, I don't think their existence in a mineralised quartz vein, amongst some larger polymetallic melt droplets, is evidence that they formed early. It's just evidence that they existed at the same place and time. It is plausible that they could have formed a long way away and been carried to the site of deposition, but it's probably more likely that they formed as the quartz hosting them crystallised, given that this would have been in response to changing T, P and/or X conditions (X being chemistry).

Agreed. By “early”, we meant at some point between the source and the deposition site, not necessarily in the causative intrusion (lines 237–239). In this way, part of the micro- or nano-sized PMIs must have formed “early” given the solubility limitations expected for a single hydrothermal fluid injection responsible for a narrow, Au-Ag over-enriched, bonanza-type ore. We kindly refer the Reviewer to a previous response (“If you're looking at inclusions trapped in mineralised...”).

You could delete the over interpretation from here and leave the overall point til later after you've discussed physical carriage.

Done (lines 139–142).

Fig 3a, b: In a, move the numbers down onto the x-axis and perhaps find a better way to indicate the size bins (why not use whole numbers to save space?).

Done.

In b, why is the critical size in the middle of the fluid ascent rate range? Would be more conservative to be at the bottom – that's also similar to the size of many of your particles, which might explain why they're there.

Agreed. However, we have modified the original Fig. 3b to present the new model.

Line 185: "...beyond which they would have sank and been lost." Hydrothermal plumbing systems are not continuous vertical pipes through which things can be carried or sink. They are tortured pathways, commonly with a significant horizontal component, that are activated by fluid-mediated deformation. Yes, small particles can be carried when the flow rate is high enough, but no, particles will not sink; they just will be too heavy to be carried along. See next comment.

Agreed. We have rewritten this whole section (lines 232–319).

General point about the usage of Stokes Law: Using the simple version of Stokes Law for this exercise is a major simplification of all hydrothermal systems, which inherently have complex structurally controlled fluid flow pathways. Commonly the structures that allow fluid transit are activated by addition of fluid from below in a dynamic environment (increased pore fluid pressure causes the rocks to break, allowing fluid transmission). The simple version of Stokes Law applies to a continuous body of fluid, such as a vertical pipe with frictionless walls, that dense particles sink through; there is no tortuosity in the migration path. Those working on particle sinking through melt networks in a solid framework of crystals have derived modifications of Stokes Law to deal with the tortuosity problem. But this still wouldn't apply directly to your case. Perhaps somebody has already attempted to numerically model carriage of dense particles in a hydrothermal fluid system?

We fully agree that fluid flow and PMIs transport occurs in a complex network of fractures, with variations in fracture aperture and rugosity. Modeling these processes through such a system would require modeling (a) fluid flow, (b) mass transport of particles, and (c) heat flow/transfer. Some approaches simplify the fracture network by an equivalent representation using the cubic law for fluid flow and assuming that the fracture network can be represented as a continuum. Such an approach requires extensive work (e.g., Liu et al., 2025) and is beyond the scope of our work.

However, to show the feasibility of PMIs transport, as a first approximation, we have modelled this network as a single tube filled with solid particles (among which are the PMIs), employing the formalism of fluidized beds (equation 1); these particles fill the fracture space up to some fraction ($1-\phi$), while the rest of the space is occupied by the fluid (ϕ). We now estimate the minimum velocity for particles within such a system/tube to be mobilized/fluidized by the flowing fluid. Flow rates would also need to overcome the settling of particles (equation 2), which now we calculate from an expression valid for a wide range of Reynolds numbers (Fig. 3b), including turbulence. We also estimate the minimum pressure gradient (of course, the pressure gradient is independent of pressure magnitude) needed to fluidize the granular bed along a given length (Fig. 3c) of an existing fracture network (i.e., fault systems at Natividad). We emphasize that tortuosity within the fracture network would increase the pressure gradient needed to mobilize the bed (lines 190–192).

Our new approach shows that PMIs, especially nanos, are readily fluidized ($<10^{-5}$ m/s) and settle slowly ($<10^{-2}$ m/s; Fig. 3b). These results suggest that if particles settle at some point due to fluid stagnation under an impermeable barrier, they could be easily fluidized as more fluid is added from below, and ascent is resumed. Therefore, hydrothermal solutions can potentially fluidize and drag

nanoPMIs, even in tortuous pathways across the uppermost continental crust, considering ascent rates for hydrothermal fluids between 10^0 and 10^{-2} m/s (Delaney et al., 1987; Okamoto and Tsuchiya, 2009; Dissanayake et al., 2014). This velocity range is conservative, given that some hydrothermal plumes reach up to 6.2 m/s (Dissanayake et al., 2014).

Although our model may be regarded as simplistic, previous authors have employed a similar approach to explain the mechanical transport of dense particles within hydrothermal fluids (e.g., Oliver et al., 2006; Okamoto and Tsuchiya, 2009). For example, Oliver et al. (2006) modelled the carriage of rock fragments up to 10 m in size by magmatic-derived fluids over ~10 km, which resulted in “fluidized breccias” at the Cloncurry district, Australia. We partly based our approach on theirs, as they also calculated fluidization and particle settling velocities as well as the minimum pressure gradient necessary to maintain fluidization conditions along distances up to 10 km. Moreover, Okamoto and Tsuchiya (2009) modelled settling velocities of quartz crystals up to 500 μm long during fluid ascent within vein-forming fractures in the Sambagawa Belt, Japan. They also used an approach similar to Oliver’s (and ours), although they simplified their particle settling model by using the Stoke’s law ($\text{Re} < 1$).

Currently, Figure 3b and the discussion surrounding it are largely irrelevant because of this over simplification. The discussion on whether a given size particle can be transported is not based on appropriate modeling. Are the fluid ascent rates given an average of the ascent rate through a broad section of crust, or are they local ascent rates applicable to a particular structure?

Ascent rates were taken from authors who measured or calculated ascent rates of hydrothermal fluids in black smokers (Delaney et al., 1987; Dissanayake et al., 2014) and vein-forming fractures (Okamoto and Tsuchiya, 2009). We took these ascent rates as average values from the PMI source(s) to the deposition site. Our aim was not to model the transport of PMIs across a broad section of the crust, given that intermediate-sulfidation epithermal deposits are usually emplaced less than 4 km from their causative intrusion (Camprubi and Albinson, 2007; Sillitoe, 2010; Wilkinson et al., 2013), which is the uppermost section of the crust. Nevertheless, it is worth noting that some authors (Schettino et al., 2023) have shown that polymineral inclusions (solidified melts) can be physically transported by ascending silicate melts from the mantle to the upper continental crust.

One possible way to deal with this problem is to acknowledge the difficulty of modeling a complex system, show that the simple Stokes Law modeling finds that sub-micron particles settle at rates 3 orders of magnitude slower than estimated fluid ascent rates (Fig 3b), and therefore suggest that nanodroplets are likely to be easily transported long distances in hydrothermal systems (not the cautious tone of the text). The other way to deal with the problem is to numerically model the complex system, which is perhaps what’s needed for a Nature Comms paper.

Agreed.

Lines 185-189: “Recent studies 8,9,15,27 have shown that shallow boiling, cooling, and mixing with external fluids (such as meteoric water or seawater) promote the coalescence of nano-sized materials in hydrothermal fluids. These processes are characteristic of the shallow epithermal environment 28,29, suggesting that some polymetallic melts from El Hilo should have originated outside this environment.” Sure, coalescence is likely promoted in that environment, but this doesn’t require that the particles formed outside that environment. Given the dramatic changes in P, T and X, it’s more likely that they formed when the fluids experienced boiling, cooling, mixing. I agree that it’s plausible that some of them formed prior to reaching the site of deposition.

We agree with Dr. Tomkins that placing the exact site of nanomelt formation is challenging. For this reason, we have suggested in lines 237–239 and 309–310 that nanomelts could have formed anywhere between the magmatic source and the deposition site. Nevertheless, the magmatic input to ores is shown by sulfides and sulfosalts with $\delta^{34}\text{S}_{\text{VCDT}}$ between -3.2‰ and -0.3‰ (Cano et al., 2024), which

matches the -3% to 3% range of mantellic S (Chaussidon and Lorand, 1990). The epithermal environment may have a vertical extend of ~ 2 km in similar geologic settings (e.g., Camprubí and Albinson, 2007), which gives plenty of space for nanomelts to form and grow during fluid transit as a result of boiling, fluid-rock interactions, and conductive heat loss. Regardless of the site of formation, very high precious metal contents at El Hilo added to the abundance of Au-Ag enriched PMIs suggest that some nanomelts were likely physically transported from a certain source (magmatic or epithermal) to the final site—bonanzas are hard to explain by “regular” precipitation from solution (Au-Ag solubilities in the order of ppbs; e.g., Seward et al., 2014; Pearce et al., 2015; Wagner et al., 2016; Liu et al., 2019).

We cannot discard any potential source of nanomelts in the magmatic-hydrothermal system, especially considering that LMCE-bearing mineral assemblages (i.e., former nanomelts) have been reported in porphyry deposits that represent the roots of some types of epithermal deposits (e.g., Elatsite in Bulgaria; González-Jiménez et al., 2021) and constitute a good proxy for the likelihood of a magmatic source for nanomelts.

CITED REFERENCES

- Camprubí, A., and Albinson, T., 2007, Epithermal deposits in México - Update of current knowledge and an empirical reclassification: Special Paper of the Geological Society of America, v. 422, p. 377–415, doi:10.1130/2007.2422(14).
- Cano, N., Camprubí, A., Proenza, J.A., and González-Partida, E., 2024, Genesis and evolution of the Natividad Au-Ag(-Ge) epithermal deposit in southern Mexico: Unravelling the final chapters of the Sierra Madre del Sur metallogeny: Ore Geology Reviews, doi:https://doi.org/10.1016/j.oregeorev.2024.105979.
- Chaussidon, M., and Lorand, J.P., 1990, Sulphur isotope composition of orogenic spinel lherzolite massifs from Ariège (North-Eastern Pyrenees, France): An ion microprobe study: *Geochimica et Cosmochimica Acta*, v. 54, p. 2835–2846, doi:10.1016/0016-7037(90)90018-G.
- Delaney, J.R., Mogk, D.W., and Mottl, M.J., 1987, Quartz-cemented breccias from the Mid- Atlantic Ridge: samples of a high-salinity hydrothermal upflow zone.: *Journal of Geophysical Research*, v. 92, p. 9175–9192, doi:10.1029/JB092iB09p09175.
- Dissanayake, A.L., Yapa, P.D., and Nakata, K., 2014, Simulation of hydrothermal vents in the Izena Cauldron, Mid Okinawa trough, Japan and other Pacific locations: *Journal of Hydro-Environment Research*, v. 8, p. 343–357, doi:10.1016/j.jher.2014.05.003.
- González-Jiménez, J.M., Piña, R., Kerestedjian, T.N., Gervilla, F., Borrajo, I., Pablo, J.F. de, Proenza, J.A., Tornos, F., Roqué, J., and Nieto, F., 2021, Mechanisms for Pd–Au enrichment in porphyry-epithermal ores of the Elatsite deposit, Bulgaria: *Journal of Geochemical Exploration*, v. 220, p. 106664, doi:10.1016/j.gexplo.2020.106664.
- Guisbiers, G., Mendoza-Cruz, R., Bazán-Díaz, L., Velázquez-Salazar, J.J., Mendoza-Perez, R., Robledo-Torres, J.A., Rodríguez-Lopez, J.L., Montejano-Carrizales, J.M., Whetten, R.L., and José-Yacamán, M., 2016, Electrum, the gold-silver alloy, from the bulk scale to the nanoscale: Synthesis, properties, and segregation rules: *ACS Nano*, v. 10, p. 188–198, doi:10.1021/acsnano.5b05755.
- Liu, W., Chen, M., Yang, Y., Mei, Y., Etschmann, B., Brugger, J., and Johannessen, B., 2019, Colloidal gold in sulphur and citrate-bearing hydrothermal fluids: An experimental study: *Ore Geology Reviews*, v. 114, p. 103142, doi:10.1016/j.oregeorev.2019.103142.
- Liu, B., Kumar, D., and Ghassemi, A., 2025, Modeling proppant transport and settlement in 3D fracture networks in geothermal reservoirs: *Geothermics*, v. 125, doi:10.1016/j.geothermics.2024.103176.
- Okamoto, A., and Tsuchiya, N., 2009, Velocity of vertical fluid ascent within vein-forming fractures: *Geology*, v. 37, p. 563–566, doi:10.1130/G25680A.1.

- Oliver, N.H.S., Rubenach, M.J., Fu, B., Baker, T., and Blenkinsop, T.G., 2006, Granite-related overpressure and volatile release in the mid crust: fluidized breccias from the Cloncurry District, Australia: *Geofluids*, v. 6, p. 346–358, doi:10.1111/j.1468-8123.2006.00155.x.
- Pearce, M.A., White, A.J.R., Fisher, L.A., Hough, R.M., and Cleverley, J.S., 2015, Gold deposition caused by carbonation of biotite during late-stage fluid flow: *Lithos*, v. 239, p. 114–127, doi:10.1016/j.lithos.2015.10.010.
- Schettino, E., González-Jiménez, J.M., Marchesi, C., Palozza, F., Blanco-Quintero, I.F., Gervilla, F., Braga, R., Garrido, C.J., and Fiorentini, M., 2023, Mantle-to-crust metal transfer by nanomelts: *Communications Earth and Environment*, v. 4, p. 1–9, doi:10.1038/s43247-023-00918-y.
- Seward, T.M., Williams-Jones, A.E., and Migdisov, A.A., 2014, The chemistry of metal transport and deposition by ore-forming hydrothermal fluids, in Holland, H. and Turekian, K.K. eds., *Treatise on Geochemistry: Second Edition*, Oxford, Elsevier Ltd., p. 29–57, doi:10.1016/B978-0-08-095975-7.01102-5.
- Sillitoe, R.H., 2010, Porphyry copper systems: *Economic Geology*, v. 105, p. 3–41, doi:10.2113/gsecongeo.105.1.3.
- Wagner, T., Fusswinkel, T., Wälle, M., and Heinrich, C., 2016, Microanalysis of fluid inclusions in crustal hydrothermal systems using laser ablation methods: *Elements*, v. 12, p. 323–328.
- Wilkinson, J.J., Simmons, S.F., and Stoffell, B., 2013, How metalliferous brines line Mexican epithermal veins with silver: *Scientific Reports*, v. 3, p. 1–7, doi:10.1038/srep02057.

Response to the comments by Reviewer 3
(Reviewer's comments are italicized and our responses are in blue)

This manuscript introduced a new mechanism of enrichment of noble metals in magmatic-hydrothermal ore forming systems. The authors propose that nano-micro sized melts may be present and suspend in hydrothermal fluids and be transported by fluids to form mineralization. Evidence provided includes SEM, EDS, TEM and EPMA analyses, etc. The idea reads interesting with relevant data and observations. However, I have some concerns related to whether the melts could be stable in the fluids at the T-P-salinity conditions as discussed, as well as if the calculation of the fluids' capability to transport and maintain melts suspension is accurate enough. Please see below:

We thank Reviewer 3 for taking the time to revise our manuscript and for the positive feedback about the novelty of our investigation. Their suggestions have considerably improved it. We have responded to their comments point-by-point below.

1. It is important to consider the chemical stability of the noble metal melts in the hydrothermal fluids with relatively high T and salinity. At conditions of El Hilo fluids, e.g., 100-400°C, up to 20wt% NaCl, the fluids may dissolve significant amounts of metals, in particular when considering that chloride complexes are thought to be important species for Au, Ag, and PGEs in Cl-dominated fluids and contribute to their hydrothermal transportation. Hence, it is important to provide constraints on if the nano-micro metal melts can be chemically stable as well in such fluids. I suggest the authors to provide thermodynamic calculations of the relevant metals and if possible, the fluid composition measurement of the fluid inclusions to demonstrate what the solubility of metals is.

We agree with Reviewer's comment. At fluid conditions of El Hilo, part of the Au and Ag load was probably transported in solution as AgCl_2^- and/or AuHS_2^- . However, available solubility models for hydrothermal fluids at 10 wt% NaCl equiv., pH ~5, and 350 °C (Kouzmanov and Pokrovski, 2012)—which are conditions similar to those of El Hilo fluids—yield solubility limits of ~0.1 ppm Au and 50 ppm Ag. These data agree with calculations by other authors that suggest Au solubilities on the order of ppbs (e.g., Seward et al., 2014; Pearce et al., 2015; Wagner et al., 2016; Liu et al., 2019). Considering this, the dissolution of large amounts of Au and Ag present at El Hilo (in the wt% scale) would have required large volumes of hydrothermal fluids, which are not reflected in the extent of quartz veining and/or alteration haloes when compared to neighboring low-grade or unmineralized veins (lines 117–125). Therefore, this hydrothermal pulse (El Hilo bonanza represents a single fluid injection, as it lacks any laminated textures) could have not possibly dissolved all precious metals contained in suspended materials (PMIs), which probably hindered their dissolution and consequently favored their chemical stability with the media. Nevertheless, we think that metal-rich nanomelts were not inert because they could have potentially scavenged Au and Ag from the solutions (e.g., Tooth et al., 2011), and hence the very high Au and Ag contents of PMIs (2 wt% Au and 31 wt% Ag).

As for the second part of the comment, previous authors have performed thermodynamic calculations and solubility models for metals in hydrothermal fluids at different temperature, salinity, and pH ranges (e.g., Kouzmanov and Pokrovski, 2012; Seward et al., 2014; Pearce et al., 2015; Liu et al., 2019). Their results have shown that Au and Ag solubilities are on the order of ppbs, which would not account for the formation of precious metal bonanzas (McLeish et al., 2021; Petrella et al., 2022), as already noted. In addition, we did not consider microchemical analyses on fluid inclusions because those measurements would not provide trustworthy constraints on metal solubility given the occurrence of suspended metal-rich nanomaterials. Analyzing fluid inclusions (e.g., Fig. 1c) by, for example, LA-ICP-MS would yield unreliable Au and Ag contents (Au-Ag in solution + Au-Ag in suspension).

2. The calculation of the capability of upwelling fluids to mechanically transport particles is based on the assumption of the fluids of a fixed density of 840kg/m³ at 340°C and 300 bar. This was then applied to calculate the settling velocity of fluids with various salinities. However, it is important to note that the salinity has a significant influence on the fluid density as well. I have done some calculations myself: for fluid at 340°C, 300bar, the density of fluids for 0m, 2m, and 4m NaCl would be 670, 793.7, and 879.14kg/m³ respectively. There is a big variation there, and what is the influence of such on the settling velocity? The authors state that the salinity has an insignificant impact on the settling velocity because they ignored salinity's impact on density. I suggest the authors to revise their calculations and provide a more accurate estimate.

We fully agree with Reviewer's comment and are thankful for performing their own calculations. After calculating fluidization and settling velocities of particles carried by fluid with densities of 670, 793.7, and 879.14 kg/m³, we confirmed that density (and hence salinity) has no remarkable impact on the results (see figure below and Table S3). Thus, we have performed new calculations of fluidization conditions and settling velocities at a fixed density (837 kg/m³; average density of fluid inclusions from El Hilo, see Table S3). Instead, we noted that other variables, such as the porosity of the bed of suspended particles (fluid reservoir with melt droplets) or the fluid regime (Reynolds number), have a more significant effect on fluidization and settling velocities than fluid density. For these reasons, our new models contemplate different porosities (0.3, 0.5, and 0.7) and fluid regimes (Re = 1, 10, and 100).

Calculation of fluidization ($\phi = 0.3, 0.5, \text{ and } 0.7$) and settling ($Re = 1, 10, \text{ and } 100$) velocities of particles within fluids with contrasting densities: 670 kg/m³ (0 wt% NaCl_{eq}), 794 kg/m³ (10 wt% NaCl_{eq}), and 840 kg/m³ (19 wt% NaCl_{eq}). Density variations are indicated by the three lines for each ϕ and Re value; for example: fluidization velocities are represented by gray (19 wt% NaCl_{eq}), orange (10 wt% NaCl_{eq}), and blue (0 wt% NaCl_{eq}) lines at $\phi = 0.3$. This plot shows that density has a negligible effect on fluidization velocities of particles with any given size, and this effect is even lower on the settling velocities (all three lines for $Re = 1, 10, \text{ and } 100$ lie on top of each other).

CITED REFERENCES

Kouzmanov, K., and Pokrovski, G.S., 2012, Hydrothermal Controls on Metal Distribution in Porphyry Cu (-Mo-Au) Systems: Society of Economic Geologists, Special publication, v. 16, p. 573–618, doi:10.5382/sp.16.22.

- Liu, W., Chen, M., Yang, Y., Mei, Y., Etschmann, B., Brugger, J., and Johannessen, B., 2019, Colloidal gold in sulphur and citrate-bearing hydrothermal fluids: An experimental study: *Ore Geology Reviews*, v. 114, p. 103142, doi:10.1016/j.oregeorev.2019.103142.
- McLeish, D.F., Williams-Jones, A.E., Vasyukova, O. V., Clark, J.R., and Board, W.S., 2021, Colloidal transport and flocculation are the cause of the hyperenrichment of gold in nature: *Proceedings of the National Academy of Sciences of the United States of America*, v. 118, p. 1–6, doi:10.1073/pnas.2100689118.
- Pearce, M.A., White, A.J.R., Fisher, L.A., Hough, R.M., and Cleverley, J.S., 2015, Gold deposition caused by carbonation of biotite during late-stage fluid flow: *Lithos*, v. 239, p. 114–127, doi:10.1016/j.lithos.2015.10.010.
- Petrella, L., Thébaud, N., Fougereuse, D., Tattitch, B., Martin, L., Turner, S., Suvorova, A., and Gain, S., 2022, Nanoparticle suspensions from carbon-rich fluid make high-grade gold deposits: *Nature Communications*, v. 13, p. 1–9, doi:10.1038/s41467-022-31447-5.
- Seward, T.M., Williams-Jones, A.E., and Migdisov, A.A., 2014, The chemistry of metal transport and deposition by ore-forming hydrothermal fluids, in Holland, H. and Turekian, K.K. eds., *Treatise on Geochemistry: Second Edition*, Oxford, Elsevier Ltd., p. 29–57, doi:10.1016/B978-0-08-095975-7.01102-5.
- Tooth, B., Ciobanu, C.L., Green, L., and Neill, B.O., 2011, Bi-melt formation and gold scavenging from hydrothermal fluids : An experimental study: *Geochimica et Cosmochimica Acta*, v. 75, p. 5423–5443, doi:10.1016/j.gca.2011.07.020.
- Wagner, T., Fusswinkel, T., Wälle, M., and Heinrich, C., 2016, Microanalysis of fluid inclusions in crustal hydrothermal systems using laser ablation methods: *Elements*, v. 12, p. 323–328..

Response to the comments on the manuscript:

A transient non-soluble noble metal transport in hydrothermal ore systems

Néstor Cano, José M. González-Jiménez, Antoni Camprubí, Eric Morales-Casique, and Eduardo González-Partida

Response to the comments by Reviewer 1

(Reviewer's comments are italicized and our responses are in blue)

I have reviewed the revised manuscript and am satisfied with the authors' responses to my comments. I believe the points raised during the initial review have been addressed thoroughly and with appropriate consideration. The revisions have significantly improved the clarity and strength of the manuscript. I have no further comments and consider the manuscript, in its current form, to be suitable for publication.

We thank Reviewer 1 for taking the time to rereview our manuscript and for their positive feedback. Certainly, their comments helped us to improve the paper considerably.

Summary of other concerns from this revision, in light of Reviewer #3's comments:

Reviewer #3 highlights an important point about the need to provide the density of the PMIs. This is critical since these metals are very dense, which affects their ability to be transported upwards. The authors could explore whether silica gel (as described in McNab et al, 2024 Coprecipitation of amorphous silica and gold nanoparticles contributes to gold hyperenrichment- GEOLOGY) might carry these metals, potentially mitigating the density issue. However, the gel would also need continuous porosity for transport.

We agree that the density of PMIs is a critical factor influencing their ability to remain suspended in the fluid during ascent and have incorporated this consideration into our model (Fig. 3b). First, we estimated the density of PMIs based on the areal distribution of mineral species and their ideal densities (sourced from mindat.org). Using data from 100 PMIs, we calculated an average density of $6.3 \pm 1.3 \text{ g/cm}^3$ (1σ) (see Supplementary Data S2, sheet "PMI composition and density," and lines 373–380).

Next, using this average density of 6.3 g/cm^3 , we computed the fluidization and settling velocities of micro- and nano-sized materials (lines 390–394 and Supplementary Data S3). Our results demonstrate that the mineralizing fluid could indeed transport such high-density materials upwards—despite the significant density difference (0.8 g/cm^3 for the fluid vs. 6.3 g/cm^3 for the melt)—due to the small size of the materials (generally $<5 \mu\text{m}$; Fig. 3a–b).

While silica gels were likely present, as suggested by the occurrence of jigsaw quartz at El Hilo, their primary role was associated with *in-situ* boiling-induced deposition of PMIs rather than facilitating the long-distance transport of nano- and micromaterials (Cano et al., 2023).

It is unclear whether metals are transported as a melt suspended in fluid or as solid nano- or microparticles (example line 257). If transported as a melt, what processes cause solidification at the site of mineralisation? The explanation provided—boiling leads to deposition—is vague and there is confusion between the coalescence of melt vs solid particles. Both references cited here 16 and 35 are describing the coalescence of solid metallic nanomaterial not melts. It is likely that the coalescence mechanism between a solid particle and melt differ and this point should be addressed in the discussion.

Agreed. Our SEM and TEM observations indicate that, prior to deposition, metals were transported as a fluid-mediated melt enriched in low-melting-point chalcophile elements (e.g., As, Sb, Se; Fig. 4a). However, *in-situ* processes at the deposition site, such as boiling and conductive cooling, were likely to have depressed the system's temperature, thus triggering the solidification of metal-rich sulfide-sulfosalt melts (referred to hereafter simply as “metal-rich melts”) after their trapping in quartz (lines 233–240). Solidified metal-rich melts are equivalent to polymineral inclusions or PMIs (lines 53–54, 145–147, 164–165, and 151–152). We have modified several parts of the text to clarify this issue.

Regarding the coalescence of melts vs. solid particles, the chemical similarity observed between nano- and micron-scale polymineral inclusions (lines 229–232; Figs. 1e–j and 2b–m) supports the coalescence of (nano)melts rather than solids. Since references 16 and 35 address the coalescence of solid nanomaterials from colloidal suspensions, we have reformulated lines 236–237 to distinguish our findings from these cases.

There are some inconsistencies in the description of the material being transported in the proposed model. For instance, in lines 126 and 135, the authors refer to a “metal-rich melt,” while later, in lines 246 and 263, they describe “sulfide melts” or “metal-rich sulfide nano melts.” This distinction is significant and needs to be clarified.

Agreed. Considering Reviewer 1’s previous comments on this issue and the average composition of PMIs (61.3 wt% Ag, 2.4 wt% Au, 4.6 wt% Sb, 2.3 wt% As, 15.5 wt% S, 1.8 wt% Se, 6.0 wt% Cu, and 3.9 wt% Fe; see Supplementary Data S2), we have chosen to refer to these as “metal-rich sulfide-sulfosalt melts” (although we used “metal-rich melts” throughout the text for brevity; lines 135–136). This terminology has been consistently applied.

In L244, the manuscript states that precious metals avoid dissolution in hydrothermal fluids, which is likely due to their low solubility but it is not the case for sulfur that has a high solubility in these systems. Then how does S remain in the melt phase rather than transitioning to the fluid phase? The stability of a sulfide melt under these conditions requires further explanation, as sulfide melts are typically associated with magmatic rather than aqueous environments.

The authors also reference the Bi collector model, where precious metals partition into a Bi melt phase in an aqueous system. While this is a well-documented process, the manuscript instead describes a sulfide melt, which may behave differently in such an environment. The stability of a sulfide melt under hydrothermal conditions, particularly its ability to persist and transport metals, needs to be addressed further in my opinion.

Additionally, if the melt is described as “metal-rich,” does it imply a sulfide-dominated composition or one dominated by precious metals with low solubility? These two scenarios could have distinct implications for melt stability, transport properties, and deposition mechanisms. Clear distinctions between these terms and their respective roles in the proposed model would help resolve potential confusion and strengthen the study’s argument.

We agree with these comments. The studied solidified metal-rich melts (i.e., PMIs) are predominantly composed of metals (primarily Ag, ~60 wt.%) and metalloids, together accounting for ~85 wt.%, with ~15 wt.% S (Supplementary Data S2). With our data, we cannot explain how S persisted in the fluid, and there is no available literature on the fate of S from metal-dominated sulfide-sulfosalt melts in low-temperature hydrothermal solutions. Nevertheless, it is reasonable that the precious metal-dominated composition of these melts—with components such as Ag and Au exhibiting very low solubility in fluids—prevented the preferential dissolution of certain constituents (e.g., S, As, Sb).

The composition of PMIs is relatively homogeneous, and we did not observe any S-free inclusions, aside from a few electrum inclusions (see figure below and Supplementary Data S2). Therefore, we classified all PMIs as former metal-rich melts, grouping them into a single type of solidified melt. As a result, no differentiation between melt types was necessary in our model, which otherwise would

have had implications for melt stability, transport properties, and deposition mechanisms as highlighted by Reviewer 1. To maintain consistency, we have double-checked and ensured the systematic use of the terms “metal-rich sulfide-sulfosalt melt” or, simply, “metal-rich melts” throughout the text.

Whisker and box plot showing the composition of PMIs (Figure taken from the Supplementary Data S2).

Although most studies on melt-mediated mineralization have focused on Bi- and Te-rich ores (Douglas et al., 2000; Tooth et al., 2008, 2011; Jian et al., 2022, 2024), some authors have also recognized melt-suggestive features in S-bearing assemblages from low temperature (<400 °C) ore systems (Ciobanu et al., 2006; Cave et al., 2019; Sousa-Guimarães et al., 2019; Vikent’eva and Bortnikov, 2023; Cano et al., 2024). Notably, some studies have proposed that S-bearing melts may act as potential collectors of Au and Ag from fluids (Ciobanu et al., 2006; Cano et al., 2024), which agrees with our observations (lines 225–227). Supplementary Data S1 highlights several examples of such observations.

CITED REFERENCES

- Cano, N., González-Jiménez, J.M., Camprubí, A., Domínguez-Carretero, D., González-Partida, E., and Proenza, J.A., 2023, Nanomaterial accumulation in boiling brines enhances epithermal bonanzas: *Scientific Reports*, v. 13, p. 1–7, doi:10.1038/s41598-023-41756-4.
- Cano, N., González-Jiménez, J.M., Camprubí, A., Proenza, J.A., and González-Partida, E., 2024, Macro-to-nanoscale investigation unlocks gold and silver enrichment by lead-bismuth metallic melts in the Switchback epithermal deposit, southern Mexico: *American Mineralogist*, v. In Press.
- Cave, B.J., Barnes, S.J., Pitcairn, I.K., Sack, P.J., Kuikka, H., Johnson, S.C., and Duran, C.J., 2019, Multi-stage precipitation and redistribution of gold, and its collection by lead-bismuth and lead immiscible liquids in a reduced-intrusion related gold system (RIRGS); Dublin Gulch, western Canada: *Ore Geology Reviews*, v. 106, p. 28–55, doi:10.1016/j.oregeorev.2019.01.010.
- Ciobanu, C.L., Cook, N.J., Damian, F., and Damian, G., 2006, Gold scavenged by bismuth melts: An example from Alpine shear-remobilizates in the Highiş Massif, Romania: *Mineralogy and Petrology*, v. 87, p. 351–384, doi:10.1007/s00710-006-0125-9.
- Douglas, N., Mavrogenes, J., Hack, A., and England, R., 2000, The liquid bismuth collector model: an alternative gold deposition mechanism, in Silbeck, C.G. and Hubble, T.C.T. eds., *Understanding planet Earth; on the starting blocks of the third millenium*, 15th Australian Geological Convention, Sydney, Geological Society of Australia, p. 135.

- Jian, W., Mao, J., Cook, N.J., Chen, L., Xie, G., Xu, J., Song, S., Hao, J., Li, R., and Liu, J., 2022, Intracrystalline migration of polymetallic Au-rich melts in multistage hydrothermal systems: example from the Xiaoqinling lode gold district, central China: *Mineralium Deposita*, v. 57, p. 147–154, doi:10.1007/s00126-021-01090-z.
- Jian, W., Mao, J., Lehmann, B., Cook, N.J., Li, J., Song, S., and Zhu, L., 2024, Hyper-enrichment of gold via quartz fracturing and growth of polymetallic melt droplets: *Geology*, v. 52, p. 411–416, doi:10.1130/G51875.1/6275576/g51875.pdf.
- Sousa-Guimarães, F., Cabral, A.R., Lehmann, B., Rios, F.J., Ávila, M.A.B., de Castro, M.P., and Queiroga, G.N., 2019, Bismuth-melt trails trapped in cassiterite–quartz veins: *Terra Nova*, v. 31, p. 358–365, doi:10.1111/ter.12391.
- Tooth, B., Brugger, J., Ciobanu, C., and Liu, W., 2008, Modeling of gold scavenging by bismuth melts coexisting with hydrothermal fluids: *Geology*, v. 36, p. 815–818, doi:10.1130/G25093A.1.
- Tooth, B., Ciobanu, C.L., Green, L., and Neill, B.O., 2011, Bi-melt formation and gold scavenging from hydrothermal fluids : An experimental study: *Geochimica et Cosmochimica Acta*, v. 75, p. 5423–5443, doi:10.1016/j.gca.2011.07.020.
- Vikent'eva, O. V., and Bortnikov, N.S., 2023, Evidence of Mineral Melting in Ores of the Svetlinsk Gold Deposit, South Urals, Russia: *Doklady Earth Sciences*, v. 513, p. 1321–1325, doi:10.1134/S1028334X2360233X.

Response to the comments by Reviewer 2 (Dr. Andrew Tomkins)
(Reviewer's comments are italicized and our responses are in blue)

I've reviewed the changes the authors have made since the previous review and I can see that they've put a lot of effort into addressing each comment. With regards to my previous comments, my main criticism was around the too-simplistic use of Stoke's Law as a way of exploring the carriage of micro- and nano-scale particles. The authors have completely redone the relevant part of the manuscript and it is now much more rigorous. As I read through this new material, I was wondering, if the particles are so easy to transport, what would make them accumulate? And then, at the end of the section (Page 14, lines 253-260), the authors discuss boiling at the site of accumulation and metal-upgrading. I guess this is a reasonable way to round out this section, and that question of nano-droplet/particle accumulation is one that the research community will continue to ponder for some time. I am happy to recommend that the article be published now.

We thank Dr. Tomkins for taking the time to rerevise our manuscript and for his positive feedback!

Response to the comments by Reviewer 3
(Reviewer's comments are italicized and our responses are in blue)

The revised manuscript provides some good improvements from their previous version. More evidence was added to support the mechanical transport of PMIs. This is good, but I think there are still some more careful justifications are needed regarding this new transport mechanism of precious metals. The manuscript should not be published before these points can be well addressed.

We thank Reviewer 3 for taking the time to rereview our manuscript and for acknowledging the improvements from its previous version.

1. The author in their reply and new calculations show that porosity has a more significant influence on fluidization and settling velocities. This is apparent. However, as the calculated porosity was 'set' to 0.3, 0.5, 0.7 in their calculation, no explanation was provided how such numbers are based on.

Agreed. Porosity affects fluidization velocity but has minimal influence on settling velocity, where the Reynolds number plays a more prominent role. As noted in lines 403–405, porosity values of 0.3, 0.5, and 0.7 were initially used to represent varying fluid (ϕ) to melt droplet+particle ($1-\phi$) ratios within fractures in the hydrothermal system. In the (re)revised version of the manuscript, we have also performed calculations for porosities of 0.9 to represent a fluid-dominated medium (lines 213–216) with some suspended metal-rich sulfide-sulfosalt melts—which probably offers a better representation of the mechanical transport mechanism being addressed. The new results for porosity at 0.9 agree with previous ones (at 0.3, 0.5, and 0.7), suggesting that micro- and nanomelt droplets could indeed have been fluidized even under very low fluid flow rates (Fig. 3b).

Well, to me, even 0.3 looks like a big overestimate of porosity by looking at their SEM of PMIs. The electrum was even more dense and less porous compared to other parts of the PMIs. As the porosity of PMIs is showing such important influence on their mechanical transport in fluids (1 to >1.5 orders of magnitude by looking at their new figure, comparing PMI porosity 0.3 to 0.7), it is necessary to provide the data on the PMI density: what is the PMI porosity in your sample?

It is important to note that porosities (ϕ) of 0.3, 0.5, 0.7, and 0.9 used in calculations are NOT for PMIs but for the fluid + suspended droplets/particles within the fractures, where the proportion of fluid is ϕ and that of droplet+particle is $1-\phi$ (lines 181–182 and 403–405). Our results suggest that PMIs were molten before their entrapment, thus they were virtually not porous.

It is not possible to reliably determine the original porosity of the “particle bed” (i.e., hydrothermal fluid + melt droplets) being fluidized. The quartz/PMI ratio observed in SEM and TEM images (Figs. 1b–d, S1) does not correspond to a specific porosity. Assuming that this ratio represents the original porosity implies that all the fluid contents in solution crystallized as quartz, which is an oversimplification. Hydrothermal fluid crystallization is rarely 100% efficient, often leaving behind a residual liquid phase. Sequential mineral precipitation, influenced by changes in temperature, fO_2 , fS_2 , and fluid composition, typically results in fractionation of the residual fluid. In open systems like the fracture-controlled Natividad deposit, such residual fluids can migrate before complete crystallization occurs.

At El Hilo, the transporting agent for PMIs was an epithermal fluid (lines 21–24). It is therefore reasonable to assume that the proportion of fluid (ϕ) was significantly greater than that of the suspended PMIs ($1-\phi$) during transit, representing a high-porosity medium (lines 213–216). Given these uncertainties and the impracticality of defining a specific porosity, we calculated fluidization velocities for a range of porosities within the fracture network (0.3, 0.5, 0.7, and 0.9) to reflect various scenarios wherein melt droplets were (or were not) transported by fluids. This approach allowed us to provide a more comprehensive representation of the natural hydrothermal system. Moreover, it is worth remembering that fluid flow across the epithermal environment does not only occur in open

spaces along the entire mineralized body: the “tails” of fractures may have rather low permeabilities, and even in the most permeable sections of the fractures, permeability is drastically reduced as minerals precipitate. The analyzed associations (banded and massive veins; Fig. 1a) belong to sections in the mineralization in which high permeabilities can be reasonably assumed. If there were mineralization types that denoted very limited permeability (i.e., veinlets or stringers), we might discuss our results otherwise.

Anyway, evaluating the behavior of fluidization velocities as a function of porosity within the fractures helped us to emphasize that metal-rich sulfide-sulfosalt droplets could have been readily fluidized, even at low flow rates and very low pressure gradients (Fig. 3b–c).

2. The question then comes to the density calculation of the PMIs. It is clear if the PMIs are too dense then they will precipitate fast, not possible at all for them to be transported by a fluid.

Yes, metal-rich sulfide-sulfosalt melts were significantly denser than the fluid phase. In earlier versions of the manuscript, we provided density estimates for PMIs based on the areal distribution of mineral species and their ideal densities (sourced from mindat.org). Using data from 100 PMIs, we calculated an average density of $6.3 \pm 1.3 \text{ g/cm}^3$ (1σ). This information is detailed in Supplementary Data S2 (sheet “PMI composition and density”) and explained in lines 373–380.

Despite the substantial density difference between the fluid (0.8 g/cm^3) and the melts (6.3 g/cm^3), our calculations demonstrate that the mineralizing fluid could effectively transport these high-density materials upwards, mainly due to their nano and micron-scale sizes (Fig. 3a–c; lines 191–197; Supplementary Data S3). Larger droplets would have not been possibly transported by the fluids, which would explain the size threshold of PMIs at El Hilo around $40 \mu\text{m}$ (Fig. 3a).

The provided density calculations of PMIs were based on mineral data (solid) and the numbers were calculated by using mixtures of such through compositional data. There are two major flaws there: (1) how different will it be for a melt mixture and a solid mixture when they have the same composition?

As detailed above, we calculated the density of PMIs based on the proportion of their constituent minerals (see Supplementary Data S2). As Reviewer 3 correctly noted, these densities represent solid PMIs, not their molten counterparts (i.e., metal-rich sulfide-sulfosalt melts). However, it is important to recognize that solid metallic minerals generally have higher densities than their liquid forms (lines 392–394). This is due to the closely packed structures of solids, which maximize packing efficiency, whereas liquids lack long-range atomic order, resulting in increased atomic spacing and volume—and thus lower density. For instance, solid Ag has a density of 10.5 g/cm^3 compared to 9.3 g/cm^3 for liquid Ag, with similar trends observed for Au, Cu, Pb, S, Se, and others.

Considering these differences, if the mineralizing fluid was capable of transporting materials with densities of 6.3 g/cm^3 (the density of solidified PMIs; lines 392–394), it would have certainly been able to carry less dense materials, such as molten PMIs (Fig. 3b). Therefore, our conclusion regarding the suspension-like transport of metal-rich melts remains well-supported and consistent with these observations.

(2) the observed particle size seems to be solid rather than a polymetallic melt, which may have significantly smaller size but higher density. If such, it will be more difficult to transport them in fluid. Our SEM and TEM analyses of PMI-rich quartz reveal textures—such as blebby shapes, curvilinear or cusped boundaries, and triple points (lines 112–114)—that suggest PMIs originated as metal-rich melts. These melts, enriched in low-melting-point chalcophile elements (LMCE; As, Sb, Se), were stable within the epithermal fluid (lines 132–147). The observed textures indicate that the metal-rich

sulfide-sulfosalt melts remained liquid during transport and even after entrapment, crystallizing only as the system cooled down (lines 237–240).

As Reviewer 3 noted, the solidified PMIs are denser and would have been more challenging to transport via ascending fluids compared to their molten counterparts. Therefore, our calculations using the density of solid PMIs (6.3 g/cm³) emphasize the capability of hydrothermal solutions to transport high-density micro- and nanomaterials, even under low flow rates (Fig. 3b). This further supports the feasibility of fluid-mediated transport of metal-rich melts in such systems (Fig. 4a).

3. Then, since the electrum are more dense and has higher density than other parts of PMIs, the author should explain better how these things with such contrasted physical properties can be hold together then a fluid is trying to flush them. It is very likely the less dense part which also have higher porosity will be effectively separated with Ag-Au electrum.

Minerals forming PMIs did not need to remain held together during transport, as they represented “homogeneous” multi-component melt droplets comprising Ag-Au-Cu-Pb(-Fe-Zn)-As-Sb-S-Se (lines 112–117 and 132–147). Sinking of denser phases, such as electrum, would have only occurred if electrum had fractionated from the metal-rich melt before its entrapment. However, electrum fractionation from the melt was unlikely, as indicated by FE-SEM and TEM images showing that micro- and nano-PMIs consist of electrum, Ag sulfosalts, and base metal sulfides (Figs. 1e–j and 2b–m). Otherwise, electrum would be systematically missing from PMIs.

4. The temperature range for these melts was assumed to be <400°C, yet, the melting point of mentioned phases (see their supplementary Data S2) are mostly much higher (eg, Ag is ~960°C, Au is 1064°C). The author need to explain better in the text how such melt can be stable in such a low T circumstances.

Agreed, but numerous experimental, thermodynamic, and empirical studies have shown that the melting points of noble metals (e.g., Ag, Au, PGEs) can be significantly lowered, even to below 400 °C, when they are associated with low-melting-point chalcophile elements such as Bi, Te, Sb, Se, As, and Pb (lines 34–44; Douglas et al., 2000; Frost et al., 2002; Tomkins et al., 2006, 2007; Tooth et al., 2008, 2011; Cave et al., 2019; Jian et al., 2022, 2024; Cano et al., 2024).

The point raised by Reviewer 3 is explained in lines 132–145: “To date, no experimental investigation fully elucidates the intricate relationships among cotectics, eutectics, and thermal boundaries within the multi-component system Ag-Au-Cu-Pb(-Fe-Zn)-As-Sb-S-Se, especially at the nano-realm. Nevertheless, the existence of metal-rich As-Sb-S-Se melts (referred to hereafter simply as “metal-rich melts”, since metals account for >70 wt%; Supplementary Data S2) is consistent with fluid temperatures of ~270–400 °C¹⁷ exceeding binary and ternary eutectic points of the simpler systems: Au-Pb (212 °C), As-S (310 °C), Au-Sb (360 °C), Ag-As-S (280 °C), Pb-Sb-S (240 °C), and Pb-As-S (305 °C)^{2,4,18}. Accordingly, the stability of metal-rich melts in low-temperature (<400 °C) hydrothermal fluids is favored in (1) complex multicomponent systems containing low-melting-point chalcophile elements (LMCE), namely As, Sb, Se, Pb, Bi, etc.^{2,5}; and (2) some metallic nanomaterials (e.g., Au-Ag alloy nanoparticles <10 nm in size), whose melting temperature decreases dramatically at the nanoscale compared with bulk counterparts^{19,20}.”

5. The impact of the proposed mechanism should be better discussed. It seems that even if the mechanism is true, it is more of a localized phenomenon and may not really explain large-scale precious metal mineralization, especially for Au. It may explain El Hilo, but how to apply this to other cases? Please provide more insights.

Agreed. Although the formation of bonanza deposits, such as El Hilo, is highly constrained in both space and time during the evolution of mineralizing systems, the processes that lead to their formation offer valuable insights into metal transport mechanisms (e.g., Saunders and Burke, 2017; McLeish et

al., 2021; Petrella et al., 2022; Cano et al., 2023; Jian et al., 2024). Our findings suggest that the El Hilo deposit formed through the accumulation of metal-rich materials that were mechanically transported by fluids from an *ex-situ* source located at a distance (Figs. 3b–c and 4a).

As discussed in the final paragraph of the discussion, several studies have documented the occurrence of Au-Ag-LMCE melts associated with ore-forming fluids at low to moderate temperatures (~100–400 °C) and salinities (<20 wt% NaCl equiv.) across various deposit types, including epithermal, skarn, and greisen (see Supplementary Data S1; Bettencourt et al., 2005; Kouzmanov and Pokrovski, 2012; Bodnar et al., 2014). These deposits are produced by mixtures of magmatic metal-rich brines and low-salinity fluids (Camprubí and Albinson, 2007; Wilkinson et al., 2013; Bodnar et al., 2014; Fekete et al., 2016; Rottier et al., 2018) and may undergo comparable evolutionary processes to those described for El Hilo (Fig. 4a). Consequently, our observations provide a potential explanation for the enrichment of noble metals in hydrothermal fluids, which could contribute to high-grade mineralization in various geological environments. We have modified lines 327–330 following Reviewer’s recommendations.

This paper does not propose nanomaterial entrainment as the primary mechanism for ore formation at the Natividad epithermal deposit. Instead, it suggests this process as a plausible explanation for exceptionally high-grade bonanza zones, such as El Hilo. The conventional understanding of metal transport in solution—via chloride or thiosulfide species—remains unchallenged. Rather, we view nanomaterial entrainment as a complementary mechanism to transport in solution, playing a key role in the formation of bonanzas. While the universal occurrence of nanomaterials in epithermal deposits is a promising area for future research, it is beyond the scope of this manuscript.

CITED REFERENCES

- Bettencourt, J.S., Leite, W.B., Goraieb, C.L., Sparrenberger, I., Bello, R.M.S., and Payolla, B.L., 2005, Sn-polymetallic greisen-type deposits associated with late-stage rapakivi granites, Brazil: Fluid inclusion and stable isotope characteristics: *Lithos*, v. 80, p. 363–386, doi:10.1016/j.lithos.2004.03.060.
- Bodnar, R.J., Lecumberri-Sanchez, P., Moncada, D., and Steele-MacInnis, M., 2014, Fluid Inclusions in Hydrothermal Ore Deposits (H. . Holland & K. K. Turekian, Eds.): Oxford, Elsevier Ltd., v. 13, 119–142 p., doi:10.1016/B978-0-08-095975-7.01105-0.
- Camprubí, A., and Albinson, T., 2007, Epithermal deposits in México - Update of current knowledge and an empirical reclassification: *Special Paper of the Geological Society of America*, v. 422, p. 377–415, doi:10.1130/2007.2422(14).
- Cano, N., González-Jiménez, J.M., Camprubí, A., Domínguez-Carretero, D., González-Partida, E., and Proenza, J.A., 2023, Nanomaterial accumulation in boiling brines enhances epithermal bonanzas: *Scientific Reports*, v. 13, p. 1–7, doi:10.1038/s41598-023-41756-4.
- Cano, N., González-Jiménez, J.M., Camprubí, A., Proenza, J.A., and González-Partida, E., 2024, Macro-to-nanoscale investigation unlocks gold and silver enrichment by lead-bismuth metallic melts in the Switchback epithermal deposit, southern Mexico: *American Mineralogist*, v. In Press.
- Cave, B.J., Barnes, S.J., Pitcairn, I.K., Sack, P.J., Kuikka, H., Johnson, S.C., and Duran, C.J., 2019, Multi-stage precipitation and redistribution of gold, and its collection by lead-bismuth and lead immiscible liquids in a reduced-intrusion related gold system (RIRGS); Dublin Gulch, western Canada: *Ore Geology Reviews*, v. 106, p. 28–55, doi:10.1016/j.oregeorev.2019.01.010.
- Douglas, N., Mavrogenes, J., Hack, A., and England, R., 2000, The liquid bismuth collector model: an alternative gold deposition mechanism, in Silbeck, C.G. and Hubble, T.C.T. eds., *Understanding planet Earth; on the standing blocks of the third millenium*, 15th Australian Geological Convention, Sydney, Geological Society of Australia, p. 135.

- Fekete, S., Weis, P., Driesner, T., Bouvier, A.S., Baumgartner, L., and Heinrich, C.A., 2016, Contrasting hydrological processes of meteoric water incursion during magmatic–hydrothermal ore deposition: An oxygen isotope study by ion microprobe: *Earth and Planetary Science Letters*, v. 451, p. 263–271, doi:10.1016/j.epsl.2016.07.009.
- Frost, B.R., Mavrogenes, J.A., and Tomkins, A.G., 2002, Partial melting of sulfide ore deposits during medium- and high-grade metamorphism: *Canadian Mineralogist*, v. 40, p. 1–18, doi:10.2113/gscanmin.40.1.1.
- Jian, W., Mao, J., Cook, N.J., Chen, L., Xie, G., Xu, J., Song, S., Hao, J., Li, R., and Liu, J., 2022, Intracrystalline migration of polymetallic Au-rich melts in multistage hydrothermal systems: example from the Xiaqingling lode gold district, central China: *Mineralium Deposita*, v. 57, p. 147–154, doi:10.1007/s00126-021-01090-z.
- Jian, W., Mao, J., Lehmann, B., Cook, N.J., Li, J., Song, S., and Zhu, L., 2024, Hyper-enrichment of gold via quartz fracturing and growth of polymetallic melt droplets: *Geology*, v. 52, p. 411–416, doi:10.1130/G51875.1/6275576/g51875.pdf.
- Kouzmanov, K., and Pokrovski, G.S., 2012, Hydrothermal Controls on Metal Distribution in Porphyry Cu (-Mo-Au) Systems: Society of Economic Geologists, Special publication, v. 16, p. 573–618, doi:10.5382/sp.16.22.
- McLeish, D.F., Williams-Jones, A.E., Vasyukova, O. V., Clark, J.R., and Board, W.S., 2021, Colloidal transport and flocculation are the cause of the hyperenrichment of gold in nature: *Proceedings of the National Academy of Sciences of the United States of America*, v. 118, p. 1–6, doi:10.1073/pnas.2100689118.
- Petrella, L., Thébaud, N., Fougereuse, D., Tattitch, B., Martin, L., Turner, S., Suvorova, A., and Gain, S., 2022, Nanoparticle suspensions from carbon-rich fluid make high-grade gold deposits: *Nature Communications*, v. 13, p. 1–9, doi:10.1038/s41467-022-31447-5.
- Rottier, B., Kouzmanov, K., Casanova, V., Wälle, M., and Fontbote, L., 2018, Cyclic dilution of magmatic metal-rich hypersaline fluids by magmatic low-salinity fluid: A major process generating the giant epithermal polymetallic deposit of cerro de pasco, Peru: *Economic Geology*, v. 113, p. 825–856, doi:10.5382/econgeo.2018.4573.
- Saunders, J.A., and Burke, M., 2017, Formation and aggregation of gold (Electrum) nanoparticles in epithermal ores: *Minerals*, v. 7, p. 1–11, doi:10.3390/min7090163.
- Tomkins, A.G., Frost, B.R., and Pattison, D.R.M., 2006, Arsenopyrite melting during metamorphism of sulfide ore deposits: *Canadian Mineralogist*, v. 44, p. 1045–1062, doi:10.2113/gscanmin.44.5.1045.
- Tomkins, A.G., Pattison, D.R.M., and Frost, B.R., 2007, On the initiation of metamorphic sulfide anatexis: *Journal of Petrology*, v. 48, p. 511–535, doi:10.1093/petrology/egl070.
- Tooth, B., Brugger, J., Ciobanu, C., and Liu, W., 2008, Modeling of gold scavenging by bismuth melts coexisting with hydrothermal fluids: *Geology*, v. 36, p. 815–818, doi:10.1130/G25093A.1.
- Tooth, B., Ciobanu, C.L., Green, L., and Neill, B.O., 2011, Bi-melt formation and gold scavenging from hydrothermal fluids : An experimental study: *Geochimica et Cosmochimica Acta*, v. 75, p. 5423–5443, doi:10.1016/j.gca.2011.07.020.
- Wilkinson, J.J., Simmons, S.F., and Stoffell, B., 2013, How metalliferous brines line Mexican epithermal veins with silver: *Scientific Reports*, v. 3, p. 1–7, doi:10.1038/srep02057.